# HOMMEXX 1.0: A Performance Portable Atmospheric Dynamical Core for the Energy Exascale Earth System Model

Luca Bertagna[1], Michael Deakin[1], Oksana Guba[1], Daniel Sunderland[1], Andrew M. Bradley[1], Irina K. Tezaur[1], Mark A. Taylor[1], and Andrew G. Salinger[1]

[1]Sandia National Laboratories, PO Box 5800, Albuquerque, NM, 87175 USA

**Correspondence:** Luca Bertagna (lbertag@sandia.gov)

**Abstract.** We present an architecture-portable and performant implementation of the atmospheric dynamical core (HOMME) of the Energy Exascale Earth System Model (E3SM). The original Fortran implementation is highly performant and scalable on conventional architectures using MPI and OpenMP. We rewrite the model in C++ and use the Kokkos library to express on-node parallelism in a largely architecture-independent implementation. Kokkos provides an abstraction of a compute node or device, layout-polymorphic multidimensional arrays, and parallel execution constructs. The new implementation achieves the same or better performance on conventional multicore computers and is portable to GPUs. We present performance data for the original and new implementations on multiple platforms, on up to 5400 compute nodes, and study several aspects of the single- and multi-node performance characteristics of the new implementation on conventional CPU, Intel Xeon Phi Knights Landing, and Nvidia V100 GPU.

## 1  Introduction

In this paper, we present the results of an effort to rewrite HOMME, a Fortran-based code for global atmosphere dynamics and transport, to a performance portable implementation in C++ (which we will call HOMMEXX), using the Kokkos library and programming model (Edwards et al., 2014) for on-node parallelism. Our definition of performance portable software, for the purposes of this paper, is a single code base that can achieve performance on par with the Fortran implementation on conventional CPU and Intel Xeon Phi Knights Landing (KNL) HPC architectures while also achieving good performance on an Nvidia V100 GPU architecture.

High-Order Methods Modeling Environment (HOMME) is part of the Energy Exascale Earth System Model (E3SM Project, 2018), a globally coupled model funded by the United States Department of Energy (DOE), with version 1 released in Spring 2018; HOMME is also part of the Community Earth System Model (CESM) (Hurrell et al., 2013). The development of E3SM

is crucial for advancing climate science in directions that support DOE mission drivers over the next several decades, which generally involve energy, water, and national security issues.

The E3SM model is a multiphysics application consisting of many coupled components, including models for the atmosphere dynamics and transport (HOMME), several atmospheric physics sub-grid processes, radiative heat transfer, ocean dynamics and species transport, land surface processes, sea ice dynamics, river runoff, and land ice evolution. The need to resolve numerous physical processes over a broad range of spatial and temporal scales clearly makes E3SM an application that could benefit from effective use of exascale computing resources.

Our project was created to explore, within E3SM, the feasibility of the approach of using C++ (and Kokkos, in particular) to achieve performance on the newest HPC architectures at DOE facilities and to add agility to be prepared for subsequent changes in HPC architectures. We decided to focus on HOMME for several reasons. First, and most important, HOMME represents one of the most critical components of E3SM, typically accounting for 20-25% of run time of a fully-coupled global climate simulation. Second, it is an attractive target for refactoring, as it has a well-defined test suite. Finally, HOMME offers a high bar for comparison, as it has highly-optimized implementations for MPI and OpenMP parallelism that we could use as targets for our benchmarks (Worley et al., 2011).

Our performance improvement efforts were successful in reaching a performance portable code base. In the strong-scaled end of our studies, with relatively fewer spectral elements assigned to each node, HOMMEXX achieves better than performance parity with HOMME on conventional CPU and Intel Xeon Phi KNL, and is faster on a single Nvidia V100 GPU than it is on a single dual-socket, 32-core Intel Haswell (HSW) node. In high-workload regimes where we assign more spectral elements to a node, performance on KNL and GPU is better still: here, HOMMEXX is approximately $1.25\times$ faster than HOMME on KNL, and it is $1.2\times$ to $3.8\times$ faster on a V100 than on a HSW node. In short, HOMMEXX demonstrates that, using C++ and Kokkos, we were able to write a single code base that matched or exceeded the performance of the highly-optimized Fortran code on CPU and KNL while also achieving good performance on GPU. The source code is publicly available (Bertagna et al., 2018).

As part of the rewrite we also translated the MPI communication layers to C++, adapted them to our layouts, and made minor improvements. Scaling studies show that we did not degrade this capability from HOMME: our largest calculation was a problem of $9.8$ billion unknowns solved on $5400$ nodes and $345,600$ ranks on Cori-KNL, and was marginally faster with HOMMEXX than HOMME.

The path to reaching this result required effort. Our initial code translation of HOMME into C++ and Kokkos was indeed portable to all architectures; however, by having the original implementation to compare against, we knew that this early version was not at acceptable performance levels. The performance of HOMMEXX as presented in this paper was eventually achieved by carefully studying the most computationally-demanding routines and optimizing the code, e.g., by reducing memory movement and using explicit vectorization where possible.

This effort produced 13,000 new lines of C++ and 2,000 new lines of Fortran. Most of these lines replace code that solves the dynamical and tracer equations in HOMME. HOMMEXX uses HOMME's Fortran initialization routines.

Recent years have seen a number of efforts to create performance portable versions of existing libraries and codes, most of which have focused on portability to GPUs. Here, we mention a few of these endeavors that share some common aspects with our work.

In Dennis et al. (2017) the authors focus on optimizing HOMME for Intel x86 architectures using profiling tools and kernel extractions. By revisiting MPI communication and thread dispatch, the authors achieve better parallel scalability. Single-node performance is improved with better vectorization, data locality, and inlining, among other techniques.

Carpenter et al. (2013) present an effort to port a key part of the tracer advection routines of HOMME to GPU using CUDA Fortran, with good success. Norman et al. (2015) describe a similar effort using OpenACC. They compare this OpenACC implementation with CUDA Fortran and CPU implementations, focusing on timings, development effort, and portability of the loop structures to other architectures.

Fu et al. (2016) describe the performance of a version of the Community Atmosphere Model (CAM) ported to run on the Sunway architecture, using OpenACC. Their porting effort shows good performance on some key kernels, including kernels for atmospheric physical processes. Fu et al. (2017) also describe the performance of CAM on the Sunway architecture, using both OpenACC and Athread (a lightweight library designed specifically for Sunway processors), on up to 41,000 nodes. In this work, Athread is shown to outperform OpenACC by a factor as high as $50\times$ on some kernels.

Spotz et al. (2015) and Demeshko et al. (2018) discuss the performance portability of finite element assembly in Albany (Salinger et al., 2016), an open-source, C++, multi-physics code, to multi/many-core architectures. They focused on the Aeras global atmosphere model within Albany, which solves equations for atmospheric dynamics similar to those in HOMME (2D shallow water model, 3D hydrostatic model). Numerical results were presented on a single code implementation running across three different multi/many-core architectures: GPU, KNL and HSW.

Another performance-portability effort in the realm of climate involves the acceleration of the Implicit-Explicit (IMEX) Non-hydrostatic Unified Model of the Atmosphere (NUMA) on manycore processors such as GPUs and KNLs (Abdi et al., 2017). Here, the authors utilized the OCCA (Open Concurrent Compute Abstraction) many-core portable parallel threading run-time library (Medina et al., 2014), which offers multiple language support. Strong scalability of their IMEX-based solver was demonstrated on the Titan supercomputer's K20 GPUs, as well as a KNL cluster.

Fuhrer et al. (2014a) present a refactoring effort for the atmospheric model COSMO (Fuhrer et al., 2017). The authors describe the strategy as well as the tools used to obtain a performance portable code. Similarly to HOMME, the original code is in Fortran, while the refactored code is in C++ and makes extensive use of generic programming and template metaprogramming. Comparison with the original implementation demonstrates an improvement in the production code performance.

Although our work shares common features with all of these efforts, it is the combination of several aspects that makes it unique. First, our approach aims to obtain a single implementation, performant on a variety of architectures. Second, except for initialization and I/O, our implementation runs *end-to-end* on the device, meaning there are no copies from device to host or from host to device. Third, the original implementation already achieves good performance on CPU and KNL, with which our implementation must achieve at least parity. Finally, we rely on Kokkos for portability, leveraging an existing cutting-edge library rather than implementing an in-house new interface.

The remainder of this paper is structured as follows. In section 2 we give an overview of HOMME, its algorithms, and computational capabilities. In section 3 we briefly present the Kokkos library and its main features, and describe the implementation process and choices we used in the development of HOMMEXX. Section 4 is dedicated to an in-depth performance study of HOMMEXX, including strong scaling, on-node performance, and comparison with the original HOMME. Finally, in section 5 we summarize what we have achieved and discuss lessons learned from our porting effort.

## 2    The HOMME Dycore

Atmospheric dynamics and tracer transport in E3SM are solved by the High-Order Methods Modeling Environment (HOMME) dynamical core. HOMME utilizes the Spectral Element Method (SEM) (Taylor, 2012), which is a member of the continuous Galerkin finite-elements schemes. The SEM is a highly competitive method for fluid dynamics applications, due to its accuracy and scalability (Maday and Patera, 1989; Dennis et al., 2005; Canuto et al., 2007; Bhanot et al., 2008).

In models of global atmospheric circulation, it is common to reformulate the governing Navier-Stokes and transport equations for a new vertical coordinate, $\eta$, and to decouple horizontal (2D) and vertical (1D) differential operators (Taylor, 2012; Dennis et al., 2012). Taylor (2012) gives the resulting equations:

$$\frac{\partial \mathbf{u}}{\partial t} + (\xi + f)\mathbf{k} \times \mathbf{u} + \nabla \left( \frac{1}{2}\mathbf{u}^2 + \Phi \right) + \dot{\eta}\frac{\partial \mathbf{u}}{\partial \eta} + \frac{RT_v}{p}\nabla p = 0, \tag{1}$$

$$\frac{\partial T}{\partial t} + \mathbf{u} \cdot \nabla T + \dot{\eta}\frac{\partial T}{\partial \eta} - \frac{RT_v}{c_p^* p}\omega = 0, \tag{2}$$

$$\frac{\partial}{\partial t}\left( \frac{\partial p}{\partial \eta} \right) + \nabla \cdot \left( \frac{\partial p}{\partial \eta}\mathbf{u} \right) + \frac{\partial}{\partial \eta}\left( \dot{\eta}\frac{\partial p}{\partial \eta} \right) = 0, \tag{3}$$

$$\frac{\partial}{\partial t}\left( \frac{\partial p}{\partial \eta}q \right) + \nabla \cdot \left( \frac{\partial p}{\partial \eta}q\mathbf{u} \right) + \frac{\partial}{\partial \eta}\left( \dot{\eta}\frac{\partial p}{\partial \eta}q \right) = 0. \tag{4}$$

Here $\mathbf{u}$ is horizontal velocity, $T$ is temperature, $T_v$ is virtual temperature, $p$ is pressure, $\Phi$ is geopotential, $\xi = \mathbf{k} \cdot \nabla \times \mathbf{u}$ is vorticity, $f$ is the Coriolis term, $R$ and $c_p^*$ are thermodynamic quantities, $\omega = Dp/Dt$ is pressure vertical velocity, $\frac{\partial p}{\partial \eta}$ is pseudo-density of moist air, and $\frac{\partial p}{\partial \eta}q$ is a tracer quantity.

In HOMME, a horizontal mesh of conforming quadrilateral elements is partitioned among MPI ranks; at the scaling limit, a rank owns one element. Vertical operations act on vertical levels and are resolved either with Eulerian or Lagrangian operators; the latter are based on remapping procedures (briefly, "remaps"). For the horizontal direction, differential operators are discretized with the SEM. A typical element structure, including the distribution of the degrees of freedom (DOFs) and vertical coordinate system, is depicted in Figure 1. In HOMME's version of the SEM, the DOFs and the quadrature points coincide. Figure 2 illustrates a typical quadrilateral mesh based on a cubed-sphere with 10 elements per cube edge ($n_e = 10$, or $3°$ resolution). On a cubed-sphere mesh, the total number of elements is $6 \cdot n_e^2$, and the number of DOFs per variable is $6 \cdot n_e^2 \cdot n_p^2 \cdot n_l$, where $n_p$ is the number of GLL points per element edge, and $n_l$ is the number of vertical levels. By default, in E3SM, $n_p = 4$ and $n_l = 72$. For the most refined mesh in this study, with $n_e = 240$ ($0.125°$ resolution), this leads to approximately $398 \times 10^6$ DOFs per variable. Figure 3 contains a visualization of total precipitable water obtained from an E3SM simulation with HOMME at $n_e = 240$ ($0.125°$ resolution).

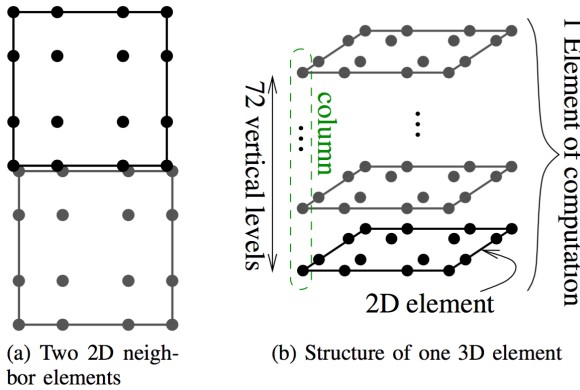

**Figure 1.** Horizontal (a) and vertical (b) structure of DOFs (marked with dots), in HOMME. In (a), DOFs along the edges of 2D elements are duplicated on each element. In (b), a 3D element consists of a stack of 72 2D elements, each with 16 GLL points.

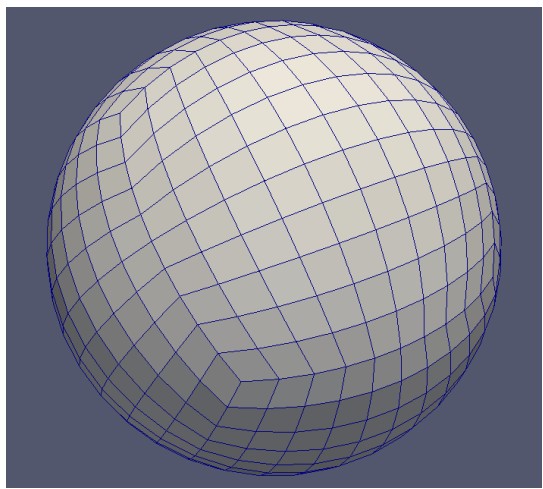

**Figure 2.** Example of a cubed-sphere quadrilateral mesh with $n_e = 10$ (3° resolution), yielding 600 elements.

The dynamical part of HOMME (briefly, "dynamics") in E3SM solves for 4 prognostic variables (pressure, temperature, and two horizontal velocity components), using a 3rd-order 5-stage explicit Runge-Kutta (RK) method and a hyperviscosity (HV) operator for grid-scale dissipation (Dennis et al., 2012). Due to timestepping restrictions, the hyperviscosity operator is subcycled 3 times per RK step. The tracer transport part (briefly, "tracers") typically solves for 40 prognostic tracers, using a 2nd-order 3-stage explicit RK method, hyperviscosity operator, and a limiter for shape preservation. The limiter is based on a local shape preserving algorithm that solves a quadratic program (Guba et al., 2014). For vertical operators, we will only consider a Lagrangian formulation, which requires a conservative and shape-preserving remapping algorithm. For the remapping algorithm, we use the piece-wise parabolic method (Colella and Woodward, 1984). A schematic of the operations

for dynamics, tracers, hyperviscosity, and remap is shown in Figure 4 as a code flow chart. Settings and parameters in our simulations match those commonly used in E3SM production runs for $1°$ horizontal resolution ($n_e = 30$).

As usual for finite element codes, SEM computations are performed in two stages: during the first stage, on-element local differential operators are computed; during the second stage, each element exchanges information with its neighbors and
computes a weighted average for the shared DOFs. Local differential operators consist of weak versions of gradient, divergence, curl, and their compositions, and are implemented as matrix-vector products.

Figure 4 shows the timestep configuration we use in this paper. There are 3 horizontal steps per one vertical remapping step. The horizontal step consists of three sequential parts: 5 RK stages for the dynamics; 3 subcycles of the hyperviscosity application for the dynamics variables; and 3 RK stages for tracers, with each stage carrying its own logic for hyperviscosity,
limiter, and MPI communication. All performance results in this paper use this configuration except in one clearly indicated case.

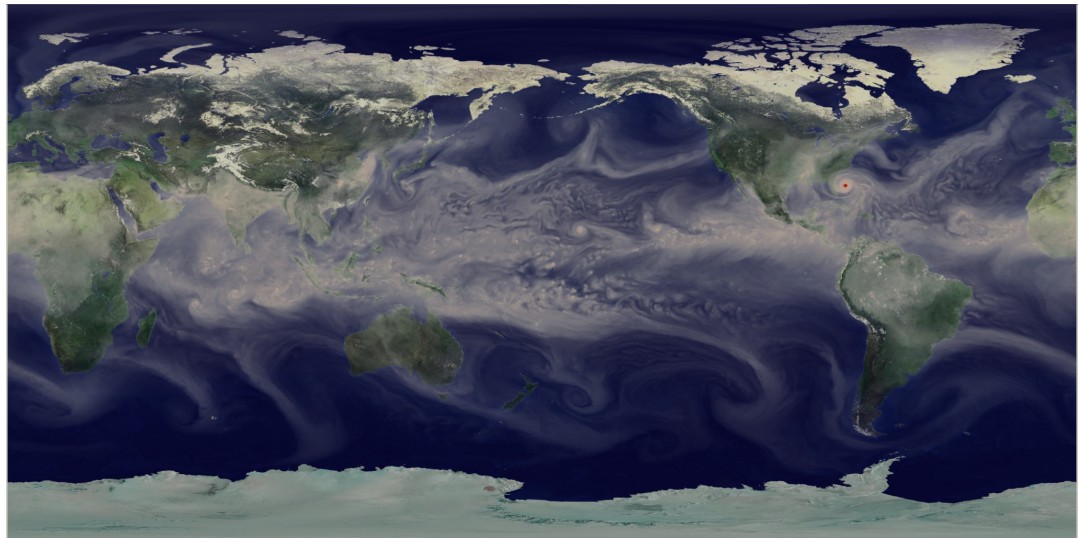

**Figure 3.** Visualization of total precipitable water, from a $0.125°$ ($n_e = 240$) E3SM simulation. The simulation used active atmosphere, land and ice component models, with prescribed ocean sea surface temperature and prescribed sea ice extent. Dynamics in HOMME alone accounts for roughly 1.6 billion degrees of freedom.

Vertical operators are not computationally intensive and are evaluated locally, while the RK stages and HV subcycles consist mostly of several calls to 2D spherical operators. Therefore, Figure 4 also highlights that a significant part of a full HOMME time step is spent in MPI communication and on-node 2D spherical operators. Due to a diagonal mass matrix in the SEM,
MPI communication involves only edges of the elements. For production runs, with $O(10^4)$–$O(10^5)$ ranks and 1–2 elements per rank, MPI cost becomes dominant relative to that of on-node computations. Previous extensive development efforts in HOMME led to very efficient MPI communication procedures. In this work, we adopt the same MPI communication patterns as in HOMME, and focus on performance and portability of on-node computations.

- • MPI exchange
- • MPI exchange for halo minmax
- ○ MPI exchange for hyperviscosity (HV)
- + limiter

**Figure 4.** HOMME timestepping configuration.

On-node computations can be parallelized through threading. On conventional CPUs, HOMME supports outer OpenMP threading over elements and inner OpenMP threading over vertical levels and tracers. The outer threads are dispatched at the driver's top-level loop in one large parallel region. Each type of threading can be turned on or off at configure time, and, if both are on, they lead to *nested* OpenMP regions. If the number of elements in a rank is at least as large as the number of hardware threads associated with the rank, then it is best to enable only the outer threads. For this reason, in our performance comparisons we configure HOMME with outer threads only, unless stated otherwise.

## 3 Overview of the implementation

### 3.1 The Kokkos library

We achieve performance portability of HOMME by using Kokkos. Kokkos is a C++11 library and programming model that enables developers to write performance portable thread-parallel codes on a wide variety of HPC architectures (Edwards et al., 2014). Kokkos is used to optimize on-node performance, allowing HPC codes to leverage their existing strategies for optimizing inter-node performance. Here, we briefly describe the key premises on which Kokkos is based.

Kokkos exposes several key compile time abstractions for parallel execution and data management which help its users to develop performance portable code. The following execution abstractions are used to describe the parallel work.

– A **kernel** is a user provided body of work that is to be executed in parallel over a collection of user defined work items. Kernels are required to be free of data dependencies, so that a kernel can be applied to the work items concurrently without an ordering.

– An **execution space** describes where a kernel should execute; for example, whether a kernel should run on the GPU or the CPU.

– An **execution pattern** describes how a kernel should run in parallel. Common execution patterns are `parallel_for`, `parallel_reduce`, and `parallel_scan`.

– The **execution policy** describes how a kernel will receive work items. A **range** execution policy is created with a pair of lower (L) and upper (U) bounds, and will invoke the kernel with an integer argument for all integers i in the interval [L, U). A **team** execution policy is created with a number of teams and number of threads per team. The strength of a team policy is that threads within a team can cooperate to perform shared work, allowing for additional levels of parallelism to be exposed within a kernel. Team policies allow users to specify up to three levels of hierarchical parallelism: over the
number of teams, the number of threads in a team, and the vector lanes of a thread.

Kokkos also provides memory abstractions for data. The main one is a multidimensional array reference which Kokkos calls `View`. `View`s have four key abstractions: (1) data type, which specifies the type of data stored in the `View`; (2) layout, which describes how the data is mapped to memory; (3) memory space, which specifies where the data lives; and (4) memory traits, which indicates how the data should be accessed. The `View` abstractions allow developers to code and verify algorithms one
time, while still leaving flexibility in the memory layout, such as transposing the underlying data layouts for CPU versus GPU architectures. Although `View`s are used to represent N-dimensional arrays, internally Kokkos `View`s store a one-dimensional array. When accessing an element (identified by a set of N indices), Kokkos performs integer arithmetic to map the input indices to a position in the internal one-dimensional array. As we will discuss in section 3, this fact impacted our design choices.

Kokkos uses C++ template metaprogramming to specify the instantiations of the execution space and data/memory abstrac-
20 tions of data objects and parallel execution constructs, to best optimize code for the specified HPC architecture. This allows users to write on-node parallel code that is portable and can achieve high performance.

HOMMEXX uses Kokkos `View`s for data management and Kokkos execution patterns for intra-MPI-rank parallelism. Hierarchical parallelism is achieved with team policies and is crucial in HOMMEXX (see section 3.3).

To give an example of Kokkos syntax, we briefly present two simple pseudocode examples. Figure 5 shows how a series of
25 tightly nested for loops can be translated to a single flattened for loop, exposing maximum parallelism. The resulting single for loop can be parallelized with Kokkos, using a simple range policy; this procedure corresponds to using OpenMP's *collapse* clause. Figure 6 shows a different scenario, in which the output of a matrix-vector multiplication (with multiple right hand sides) is used as input for a second matrix-vector multiplication (again, with multiple right hand sides). Since the second multiplication depends on the output of the first, the nested for loops cannot be flattened as in the previous case. In order to
30 use a range policy, one would have to dispatch the parallel for only on the outermost loop, which would fail to expose all the parallelism (since all the rows in each of the two multiplications can in principle be computed in parallel). With a team policy, on the other hand, threads are grouped to form teams; teams act on the same outer iteration, can share temporary variables, and can be used to further parallelize inner loops. In this example, a team is assigned to one right hand side, and each member

```
                                          Kokkos::View<double[N0][N1][N2]> a("a");
  double a[N0][N1][N2];                    Kokkos::View<double[N0][N1][N2]> b("b");
  double b[N0][N1][N2];                    Kokkos::parallel_for (
  for (int i=0; i<N0; ++i) {                 Kokkos::RangePolicy<ExecSpace>(0,N0*N1*N2),
    for (int j=0; j<N1; ++j) {   ⟹           [=](int idx) {
      for (int k=0; k<N2; ++k) {               int i = idx / (N1*N2);
        a(i,j,k) = some_function(b(i,j,k));    int j = idx / N2, k = idx % N2;
}}}                                            a(i,j,k) = some_function(b(i,j,k));
                                             });
```

**Figure 5.** Parallelization of tightly nested loops via flattening and range policy. After flattening, modular arithmetic is used by each thread to determine the portion of input and output arrays to operate on.

of the team is assigned a subset of the rows. Notice that, since the second multiplication cannot start until the first one is fully completed, a team synchronization is necessary.

Kokkos is just one of the possible ways in which one can achieve performance portability. In particular, we can identify three approaches:

– Compiler directives: In this approach, preprocessor directives expose parallelism. Included in the directives are instructions to target a specific architecture. Examples include OpenACC (OpenACC-Standard.org, 2017) and OpenMP (Chapman et al., 2007). This strategy has the advantage of requiring limited effort to adapt an existing code to run on a new architecture; but directives may not be sufficiently flexible to permit one code base to run performantly on all architectures.

– General-purpose libraries: In this approach, a third party library (written in the native language of the application) is used to interface the application with the backend architecture. The library hides architecture-specific choices from the user, while still offering handles and switches for performance optimization. The advantage of this approach is its generality (the library, together with some performance choices, can be reused across multiple applications), a mild separation of HPC concerns from scientific concerns, and a single code base. On the other hand, since the library has a general purpose,

a good understanding of HPC is still needed, in order to select the best tool from the library for the particular application. Examples include Kokkos, RAJA (Hornung and Keasler, 2014), HEMI (Harris, 2015), OCCA (Medina et al., 2014), Charm++ (Kale et al., 2008), and HPX (Kaiser et al., 2009).

   – Domain-specific languages: In this approach, the application code is written in a higher level language, which is then compiled by an intermediate source-to-source compiler, which produces a new source code, optimized for the particular 

target architecture. This code is then compiled with a standard compiler to generate the final object file. The advantage of this approach is a complete separation of HPC and science concerns: the domain scientist can write the code in a language that fully hides architecture details, focusing completely on the problem and algorithm. On the other hand, this

```
constexpr int NUM_RHS = 9;
constexpr int N = 32;
double A[N][N];
double B[N][N];
double x[NUM_RHS][N];
double y[NUM_RHS][N];
double z[NUM_RHS][N];
// Initialize arrays [...omitted...]
for (int i=0; i<NUM_RHS; ++i) {
  for (int j=0; j<N; ++j) {
    for (int k=0; k<N; ++k) {
      y[i][j] += A[j][k]*x[i][k]
    }
  }
  for (int j=0; j<N; ++j) {
    for (int k=0; k<N; ++k) {
      z[i][j] += B[j][k]*y[i][k]
    }
  }
}
```

$\Longrightarrow$

```
constexpr int NUM_RHS = 9;
constexpr int N = 32;
using Kokkos::parallel_for;
using Kokkos::parallel_reduce;
using TP = Kokkos::TeamPolicy<ExecSpace>;
Kokkos::View<double[N][N]> A("A"), B("B");
Kokkos::View<double[NUM_RHS][N]> x("x"), y("y"), z("z");
// Initialize arrays [...omitted...]
// Create policy: let Kokkos decide team size
TP policy(NUM_RHS, Kokkos::AUTO());
parallel_for(policy,
  KOKKOS_LAMBDA(TP::member_type member) {
    const int i = member.league_rank();
    parallel_for(Kokkos::TeamThreadRange(member,N),
      [=](const int& j){
        parallel_reduce(Kokkos::ThreadVectorRange(member,N),
          [=](const int& k, double& accumulator){
            accumulator += A(j,k)*x(i,k);
        },y(i,j));
    });
    member.team_barrier();
    parallel_for(Kokkos::TeamThreadRange(member,N),
      [=](const int& j){
        parallel_reduce(Kokkos::ThreadVectorRange(member,N),
          [=](const int& k, double& accumulator){
            accumulator += B(j,k)*y(i,k);
        },z(i,j));
    });
});
```

**Figure 6.** Parallelization of non-tightly nested loops via team policy. The team policy is determined by the number of outer iterations and the number of threads per team. Each team is assigned a subset of the outer iterations and, within a single outer iteration, can expose more parallelism by dividing the work among the threads in the team.

approach offers limited control to perform additional optimizations, and the scope of utilization of the tool is reduced to a limited set of target applications, for which the intermediate compiler knows how to optimize the code. Examples in this category include Stella (Mohr and Stefanovic, 2016), Grid Tools (Fuhrer et al., 2014b), and Claw (Clement et al., 2018).

When we decided to use Kokkos, we weighed the pros and cons of the approaches listed above. For instance, as suggested above, Kokkos allows a single code base to efficiently run on multiple architectures. Another reason supporting the choice of Kokkos is that it has parallel constructs that have a similar syntax to future STL parallel extensions (Open-Std.org, 2015).

### 3.2   Conversion of the existing HOMME Fortran library to C++ and Kokkos

To construct HOMMEXX, we incrementally converted the existing Fortran code to C++ with Kokkos. To ensure correctness
of the new code base, we set up a suite of correctness tests for comparing the solution computed using HOMMEXX with the solution computed using the original HOMME. For correctness tests, we enforced a *bit-for-bit* match between the solutions in correctness-testing builds on CPU and GPU platforms. For performance tests like the ones reported in this article, we enabled compiler optimizations that cause bit-for-bit differences between the codes and among architectures. In more detail, on CPU platforms and with the Intel compiler, for correctness we use the flags `-fpmodel=strict -O0`; and for performance we use
`-fpmodel=fast -O3`. On GPU platforms, for correctness, we use `-ffp-contract=off -O2` for the GNU compiler and `-fmad=false` for the CUDA nvcc compiler; for performance, we omit these flags and use `-O3`. Additionally, in the correctness-testing build for GPU, scan sums and reductions are serialized to allow bit-for-bit comparison against the CPU build; this serialization is a configuration option in HOMMEXX.

Once the testing framework was set up, we proceeded with the conversion effort. In particular, the code was divided into six
primary algorithm sets:

- differential operators on the sphere;

- an RK stage for the dynamics;

- an RK stage for the tracers, including MPI, hyperviscosity, and limiter;

- application of hyperviscosity to the dynamics states;

- vertical remap of states and tracers;

- exchange of quantities across element boundaries.

Details on the implementation of these algorithms will be given in section 3.4.

The single RK stage was the first part of the code to be converted. Initially, we separated it from the rest of the code to perform a study on the possible design choices, e.g., data layout, parallelism, and vectorization. Once the design choices were made,
we continued the conversion of the kernels, one by one, while maintaining bit-for-bit agreement with the original HOMME.

We kept the original Fortran implementation of the main entry point, as well as initialization and I/O, for the duration of the conversion process. During development, copies of data were required between C++ and Fortran code sections because the two, for performance reasons, use different memory layouts; see the next sections for details. As the conversion process progressed, we were able to push the C++ code entry points up the stack, until we were able to avoid any data movement between Fortran and C++ except in I/O and diagnostic functions.

### 3.3 Performance and optimization choices

Performance factors depend on architecture. For conventional CPU with DDR4, minimizing memory movement is the most important; for KNL, keeping a workset in high-bandwidth memory and vectorization are; for GPU, maximizing parallelism is. A performance portable design must account for all of these in one implementation.

The initial translation of HOMME into C++ with Kokkos was not as performant as the original Fortran code. This was not surprising because the Fortran code had already gone through several stages of optimization. Furthermore, Kokkos is not a black box library that automatically makes an existing code performant; effort must be put into writing efficient parallel code. We performed a sequence of optimizations that gradually improved performance. Among the different optimizations, we will now discuss the three that gave the largest performance boosts: exposing parallelism, exposing vectorization, and minimizing memory movement. We believe these three are likely to be particularly important in other applications, as well. In section 3.4.2 we discuss the implementation of the `SphereOperators` class, and present a code snippet that illustrates how these three design choices are generally used inside HOMMEXX.

**Exposing Parallelism**. Our approach to increase parallelism centered on exploiting hierarchical parallelism. HOMME has nested loops over elements, GLL points within an element, and vertical levels within an element. In HOMMEXX, these nested loops are parallelized using team policies, described in section 3.1. Since, within a kernel, each element is completely independent from the others, the outer `parallel_for` is over the elements in the MPI rank. If the kernel has to be run for several variables, as in the tracers RK stage and vertical remap, the outer `parallel_for` is dispatched over the variables in all the elements. The `parallel_for` for teams and vector lanes are dispatched over the GLL points within an element and the vertical levels, respectively. This hierarchical structure exposes parallelism with minimal bookkeeping and enables team-level synchronization. It is particularly natural for the horizontal differential operators. Such operators couple DOFs within an element. Exposing fine-grained parallelism with a simple range policy would require substantial effort. In a team policy approach, a team collectively computes a differential operator, synchronizes, and then each member uses part of the result for further calculations.

**Exposing vectorization**. This important design choice focused simultaneously on vectorization on CPU and KNL architectures and efficient threading and coalesced memory access on GPU. The two most natural choices for the primary vectorization direction are the vertical levels and the GLL points. In the former case, Kokkos' thread and vector level of parallelism would be on GLL points and vertical levels respectively, while in the latter it would be the opposite. Our choice in HOMMEXX was to go with the former, because of the following advantages compared with the latter. First, it trades vectorizing in the vertical summation regions for vectorizing in the much more frequent spherical operators. Second, while it is reasonable to change

the number of levels to be a multiple of the vector size, setting the number of GLL points, which is a perfect square, to be a multiple is too constraining. Third, given that this choice is performant, it also has the advantage of matching our approach to hierarchical parallelism on GPU.

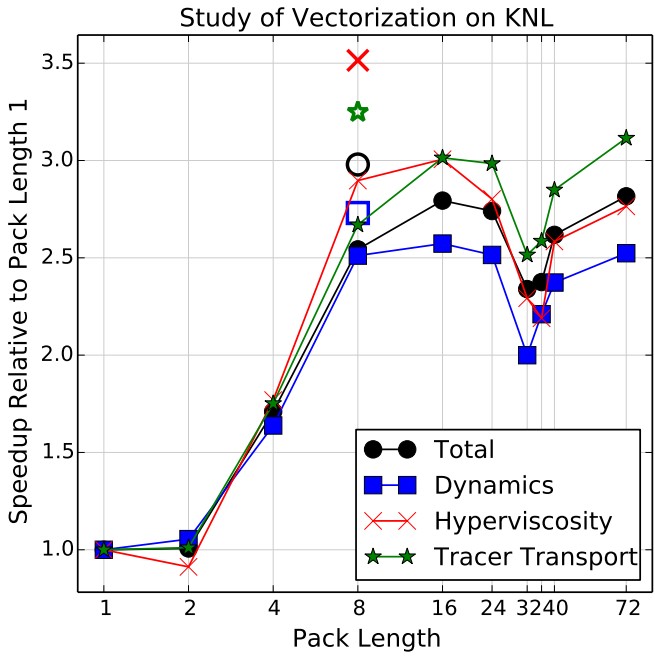

**Figure 7.** Study of vectorization performance on KNL as a function of pack length. Solid lines with solid markers correspond to auto-vectorization within each pack operator overload. Open markers correspond to explicit AVX512 intrisic calls within each operator overload.

There are several benchmarks to measure how successfully a compiler can auto-vectorize code; see, e.g., (Rajan et al.,
2015; Callahan et al., 1988). General observations in several code development efforts suggest that, roughly, well-written, but still straightforward, modern Fortran auto-vectorizes well, likely as a result of the language's built-in array arithmetic (Software.Intel.com, 2014), while C++ can be difficult to auto-vectorize well. In HOMME, very few computations have conditionals in the inner-most loops. HOMME is written with careful attention to auto-vectorization and exploits this fact.

To achieve good vectorization in HOMMEXX, we replaced the core data type, a single `double`, with a *pack*. A pack is a
group of adjacent double-precision data that supports arithmetic operations by C++ operator overload; see, e.g., Estérie et al. (2012). Packs are a key mechanism to achieve good vectorization in C++ code. Importantly, first, the pack structure has no other runtime data and, second, the pack length is a compile-time parameter. The first point, combined with data alignment rules in C/C++, implies packs of correct length for an architecture have perfect data alignment. The first and second points together imply a loop-based implementation of an arithmetic operator has a compile-time loop over perfectly aligned data.
Thus, auto-vectorization can be perfect. In the HOMMEXX implementation, we call our structure `Vector`, and it wraps an array of $N$ `doubles`. $N$ is a template parameter of the class `Vector`, which is selected at configure time depending on the available vector-unit length on the architecture. For example, on the Haswell CPU, $N$ should be a multiple of 4; on KNL, $N$

should be a multiple of 8; on GPU, $N = 1$. Analysis in a small standalone code with $N = 1$ shows that the operator overloads have no runtime impact relative to plain operators for scalars. Further analysis in a small standalone code supports that packs are crucial for C++ to achieve the same performance as well-written modern Fortran.

`Vector` provides overloads of basic arithmetic operations (+,-,*,/). In one implementation of these operators, each overload calls the corresponding vector intrinsic. A limitation of this implementation is that $N$ must be exactly the vector length of the architecture, although this limitation can be removed with more programming work. In an alternative implementation, loops are used; as explained, these auto-vectorize perfectly. This alternative implementation permits $N$ to have any size.

Figure 7 provides results of a study of the effect of pack size on HOMMEXX performance on KNL. The run configuration was 1 node, 64 ranks, 1 thread per rank, 1 rank per core, 6 elements per rank. Solid lines with solid markers correspond to auto-vectorization within each pack operator overload. Open markers correspond to the implementation with explicit AVX512 intrisic calls within each operator overload. The $x$ axis is pack length $N$; for KNL, $N$ should be a multiple of 8. The model has 72 levels, which also is an important parameter to consider when choosing $N$. Thus, $N$ should divide 72 and 8 should divide $N$. The $y$ axis shows speedup relative to pack length $N = 1$, i.e., relative to scalar operations. Data are provided for the total end-to-end time, and a breakdown among the dynamics, hyperviscosity, and tracer transport modules. Packs without intrinsic calls speed up the end-to-end time by up to approximately a factor 2.8. Intrinsic calls speed up the end-to-end time by another approximately 5%, although sections of code speed up by another approximately 15%. A pack size a multiple greater than the KNL's vector length of 8 can improve performance of the loop-based operator implementation by approximately 10%. We do not have an implementation of the intrinsics-based implementation supporting a multiple greater than 1; thus, we cannot presently conclude whether a larger multiple would be useful. In summary, a pack-based approach to vectorization provides substantial vectorization speedup on KNL, and multiple variants of the pack implementation yield a range of speedups, where this range is small relative to the overall speedup.

The use of `Vector` on CPU and KNL is efficient throughout most of the code: operations proceed independently of level, and there is no inter-level dependency. However, there are a handful of routines where at least one of these conditions does not hold. In both vertical remap and the tracer mixing ratio limiter, algorithms have a high density of conditional statements, which make vectorization inefficient. In addition, several computations require a scan over the levels. We isolate each of these from the rest of the operations to minimize non-SIMD execution.

**Minimizing memory movement**. Reducing memory movement is primarily important on conventional CPUs, which have less bandwidth than new architectures, but it improves performance on all architectures. For transient intermediate quantities, shared, reused workspace that has size proportional to number of threads is used. Persistent intermediate quantities are minimized. This proved especially important in the implementation of the horizontal differential operators (see section 3.4.2).

### 3.4 Implementation details

In this section, we highlight a few implementation details.

### 3.4.1 MPI

HOMMEXX uses the same MPI communication pattern as HOMME. The connectivity pattern between elements is computed with the original Fortran routines, and then forwarded to the C++ code. All communications are nonblocking and two-sided. The minimal number of messages per communication round is used. Each rank packs the values on the element halo in a send buffer; posts nonblocking MPI sends and receives; and unpacks the values in a receive buffer, combining them with the local data.

In HOMMEXX, this process is handled by the `BoundaryExchange` class. Several instances of this class are present, each handling the exchange of a different set of variables. Internally, these instances use, but do not own, two raw buffers: one for exchanges between elements on process (not involving MPI calls), and one for exchanges between elements across processes (requiring MPI calls). The buffers are created and handled by a single instance of a `BuffersManager` class, and are shared by the different `BoundaryExchange` object, minimizing the required buffer space.

Since CUDA-aware MPI implementations support the use of pointers on the device, there is no need for application-side host-device copies of the pack and unpack buffers for GPU builds. This can be especially beneficial if combined with GPU-Direct or NVLink technologies for fast GPU-GPU and GPU-CPU communication (Nvidia.com, 2016).

The implementation of the pack and unpack kernels is one of the few portions of HOMMEXX that differ for GPU and CPU/KNL architectures. On CPU and KNL platforms, reuse of sub-arrays (i.e., lower-dimensional slices of a multi-dimensional array) is important to minimize index arithmetic internal to Kokkos, which motivated the choice of parallelizing only on the number of local elements and exchanged variables. On GPUs, maximizing occupancy is important, which motivated the choice of parallelizing the work not only over the number of elements and exchanged variables, but also on the number of connections per element and the number of vertical levels, at the cost of more index arithmetic. Although this choice does not align with the idea of a single code base, architecture-dependent code is a very limited portion of HOMMEXX. We also use architecture-dependent code for three vertical integrals in the dynamics and the tracer transport limiter. The total number of architecture-specific lines of code is approximately 800, out of approximately 13,000 lines of C++.

### 3.4.2 Differential operators on the sphere

Discrete differential operators on the sphere are defined only in the horizontal direction, and include gradient, divergence, curl, and Laplacian. Routines implementing these operators are called extensively throughout the code, such as the RK stages for dynamics and tracers and the hyperviscosity step. In the original Fortran implementation, such routines operate on an individual vertical level. The action of an operator on a 3D field is then computed by slicing the field at each level, and passing the sub-arrays to the proper routine. As mentioned in section 2, HOMME can thread only over elements and vertical levels (or tracers) but has no support for threading over GLL points. Therefore, the body of each differential operator is serial, optimized only by means of compiler directives to unroll small loops and to vectorize array operations.

In HOMMEXX, we implemented all the differential operators on the sphere with the goal of exposing as much parallelism as possible. As mentioned in section 3.3, HOMMEXX uses three layers of hierarchical parallelism, with `parallel_for`

loops over elements (possibly times the number of tracers), GLL points, and vertical levels. The vectorization choice described in section 3.3 makes it natural to proceed in batch over levels. In our implementation, all the differential operators are computed over a whole column. Therefore, on CPU and KNL, operations are vectorized across levels, and on GPU, memory access is coalesced across levels. Some differential operators also require temporary variables, for instance, to convert a field on the sphere to a contravariant field on the cube, or vice versa. Since allocating temporaries at every function call is not performant, the `SphereOperators` class allocates its buffers at construction time. As described in section 3.3, the buffers are only large enough to accommodate all the concurrent teams.

As an example, in Figure 8 we present the implementation of the divergence operator on the sphere in HOMMEXX. The computation of the divergence involves a first loop on the $n_p \times n_p$ GLL points, to convert the input vector field on the sphere to a covariant field on a cube; then, in a second loop, the derivatives of the resulting field are computed as it is usually done in the Spectral Element Method, that is, by contracting it with the *pseudo-spectral derivative matrix* (here denoted by `dvv`). Notice that, in the second loop over the GLL points, the covariant field must be available not only at the point where the derivatives are computed, but also at all the GLL points with the same row or column index, which prevents the two loops from being fused into a single loop.

In Figure 8 we show the implementation of this operator in HOMMEXX. The routine is a method of the `SphereOperators` class, and is meant to be invoked from within a parallel region, created with a team policy. The variables `m_dinv` and `m_metdet` are related to the geometric transformation of the input field to covariant coordinates, and are stored in the `SphereOperators` object, while `vector_buf_ml` is one of the service buffers allocated inside the class. These variables are properly subviewed, in order to reduce the index arithmetic internal to Kokkos. The macro `KOKKOS_INLINE_FUNCTION` asks the compiler to inline the function call, and it informs the compiler that the function must be callable from the device. Finally, `KernelVariables` is a convenience structure holding information on the current parallel region, such as the team index, and the index of the element currently being processed. This code snippet is a good example of how the three design choices presented in section 3.3 are generally used inside HOMMEXX:

- The hierarchical structure of `parallel_for`s exposes maximum parallelism: the function is called from within a loop over the number of elements, dispatches another parallel region within the team over the number of GLL points, and finally dispatches one more parallel region within the thread vector lanes over the number of vertical levels. The last dispatch generates a parallel region only on GPU.

- The `Vector` structure (here denoted with `Scalar`) provides vectorization over a certain number of levels, depending on a configuration choice. Consequently, the number of iterations `NUM_LEV` is not necessarily the number of physical vertical levels, but rather the number of `Scalar`s into which the vertical levels are partitioned. For instance, with 72 vertical levels, and a `Vector` structure with length 8, we would have `NUM_LEV=9`. Still, the code in the routine looks exactly the same, regardless of the vector length.

– The temporary buffer `buf` is minimally sized; it is not subviewed at the current element index, like the two geometric views `m_dinv` and `m_metdet`, but instead at the current team index, so that each team can reuse its buffers at the next call to the function.

```cpp
KOKKOS_INLINE_FUNCTION void
divergence_sphere (const KernelVariables &kv,
                   const ExecViewUnmanaged<const Scalar [2][NP][NP][NUM_LEV]> v,
                   const ExecViewUnmanaged<      Scalar    [NP][NP][NUM_LEV]> div_v) const {
  const auto& Dinv = Homme::subview(m_dinv, kv.ie);
  const auto& metdet = Homme::subview(m_metdet, kv.ie);
  const auto& buf = Homme::subview(vector_buf_ml, kv.team_idx, 0);
  Kokkos::parallel_for(Kokkos::TeamThreadRange(kv.team, NP*NP), [&](const int loop_idx) {
    const int igp = loop_idx / NP, jgp = loop_idx % NP;
    Kokkos::parallel_for(Kokkos::ThreadVectorRange(kv.team, NUM_LEV), [&] (const int& ilev) {
      const auto& v0 = v(0, igp, jgp, ilev);
      const auto& v1 = v(1, igp, jgp, ilev);
      buf(0,igp,jgp,ilev) = (Dinv(0,0,igp,jgp) * v0 + Dinv(1,0,igp,jgp) * v1) * metdet(igp,jgp);
      buf(1,igp,jgp,ilev) = (Dinv(0,1,igp,jgp) * v0 + Dinv(1,1,igp,jgp) * v1) * metdet(igp,jgp);
    });
  });
  kv.team_barrier();
  Kokkos::parallel_for(Kokkos::TeamThreadRange(kv.team, NP*NP), [&](const int loop_idx) {
    const int igp = loop_idx / NP, jgp = loop_idx % NP;
    Kokkos::parallel_for(Kokkos::ThreadVectorRange(kv.team, NUM_LEV), [&] (const int& ilev) {
      Scalar dudx(0.0), dvdy(0.0);
      for (int kgp = 0; kgp < NP; ++kgp) {
        dudx += dvv(jgp,kgp) * buf(0,igp,kgp,ilev);
        dvdy += dvv(igp,kgp) * buf(1,kgp,jgp,ilev);
      }
      div_v(igp,jgp,ilev) = (dudx + dvdy) * (1.0/metdet(igp,jgp) * PhysicalConstants::rrearth);
    });
  });
  kv.team_barrier();
}
```

**Figure 8.** Implementation of the divergence operator on the sphere in HOMMEXX. This routine is meant to be called from within a parallel region dispatched with a team policy.

## 4 Performance results

Performance data were collected on a number of platforms and architectures; Table 1 summarizes these. All computations were done in double precision. All single-thread runs were done with a serial—i.e., without OpenMP—build, except as explicitly indicated. The HOMME and HOMMEXX runs for a particular comparison configuration were done in the same batch job submission to minimize the effect of external sources of noise. Jobs were short, generally not more than approximately ten minutes. Thus, comparable data points correspond to runs on the same computer nodes in a temporally constrained period of time. While noise itself inevitably remains, it affects each of the HOMME and HOMMEXX data points approximately equally.

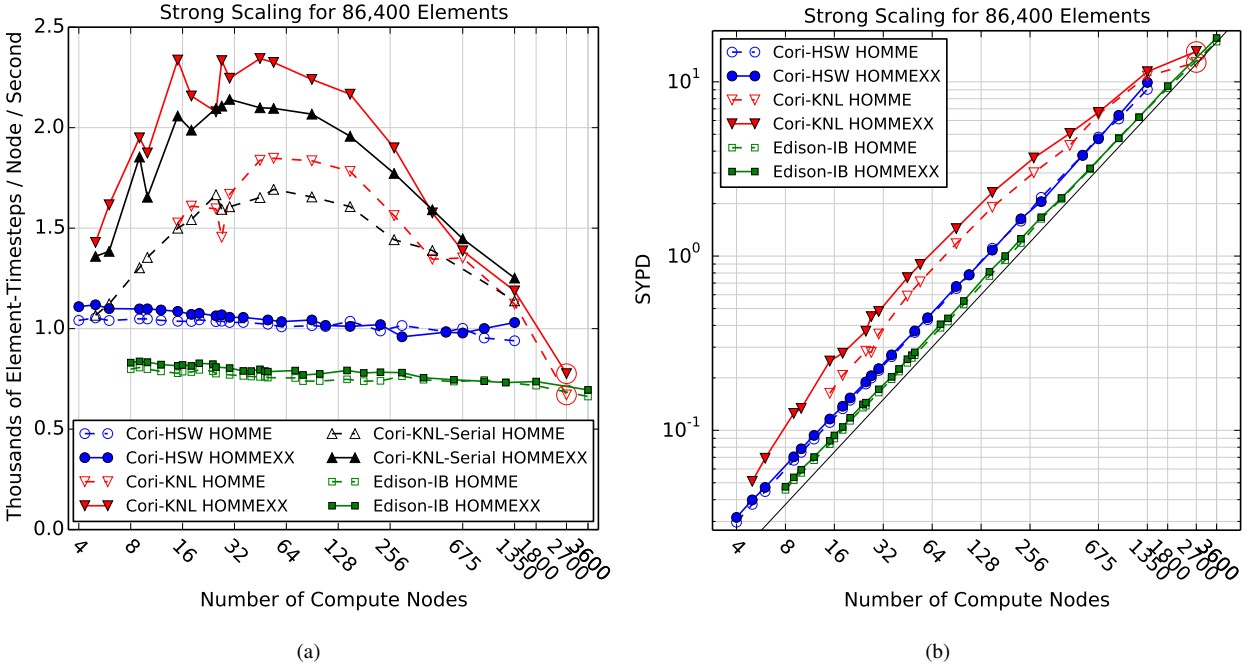

(a)                                                      (b)

**Figure 9.** Strong scaling at large scale on NERSC platforms for grid generated with $n_e = 120$ ($0.25°$ resolution). (a) The number, in thousands, of elements that can be integrated one total time step (as shown in Figure 4), per node, per second, is plotted as a function of the number of nodes. Each of the $86,400$ elements represents the work for the fourth-order spectral element extruded in 3D for 72 vertical levels, as shown in Figure 1(b), for dynamics and also advection of 40 tracers. Higher values indicate better performance. A horizontal line represents ideal strong scaling. Red circles indicate that threads are used within an element. (b) The same data are plotted in terms of Simulated Years Per wall-clock Day of compute time (SYPD). The solid black line represents ideal strong scaling.

In Figures 9, 10, and 11, the abscissa is the number of compute nodes or devices. On GPU-enabled systems, there is more than one GPU per compute node. In this case, we use the smallest number of nodes able to accommodate the given number of GPUs and count the number of GPUs rather than nodes. In Figures 12 and 13, the abscissa is the number of 3D elements.

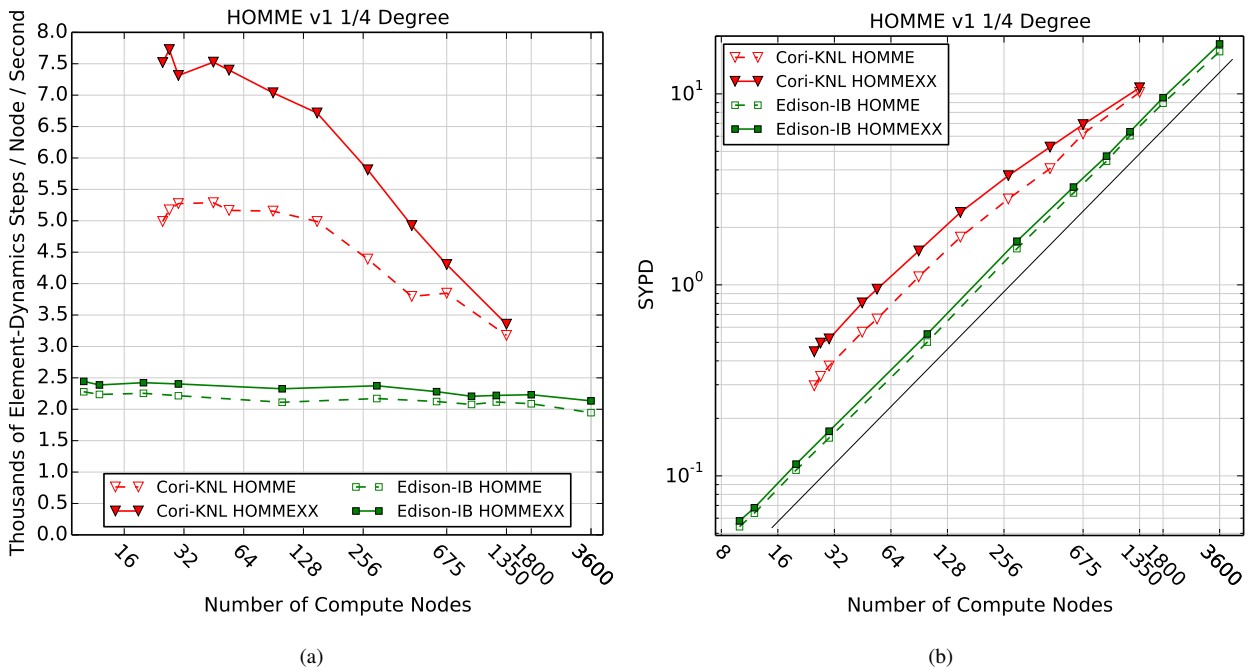

(a)  (b)

**Figure 10.** As Figure 9, but with parameters chosen according to E3SM version 1 1/4-degree model. In the version 1 model, (i) $n_e = 120$; (ii) the number of hyperviscosity subcycles is 4 instead of 3; and (iii) there are 2 horizontal steps per one vertical remapping step, instead of 3. Because of (iii), in (a) we plot average time per horizontal step plus 1/2 a vertical remapping step. To compare Figures 10 and 9, divide the $y$-axis value by 3.

The ordinate of most figures is thousands of 3D elements $\times$ total time steps computed per second, per node or GPU, where a total time step is as illustrated in Figure 4. A 3D element is the full 3D extruded domain of 72 levels as shown in Figure 1. Hence, the ordinate expresses computational efficiency; larger ordinate values indicate better performance.

In the remainder of this section, we describe results for four different data collection types: strong scaling and comparison
5   at scale, strong scaling at small scale but in more detail, single-node or device performance, and GPU kernel performance. We conclude with a discussion of power consumption.

## 4.1   Strong scaling

Figure 9 reports performance results for large runs on the NERSC Cori Haswell partition (Cori-HSW), Cori KNL partition (Cori-KNL with OpenMP, Cori-KNL-Serial with threads off), and Edison, with 86,400 elements. On Cori-HSW and Edison,
10   runs were with 1 thread, 1 rank/core. On Cori-KNL, runs were with 1 rank/core, 2 threads/core, 64 ranks/node, except for the red-circled points. These configurations are used in E3SM and thus are the best for a performance comparison study.

The red-circled points in Figure 9 were run with 1 thread/core but 2 cores/element; HOMME was configured to use inner threads and no outer threads. The serial configuration for Cori-KNL (Cori-KNL-Serial) provides a reference against which

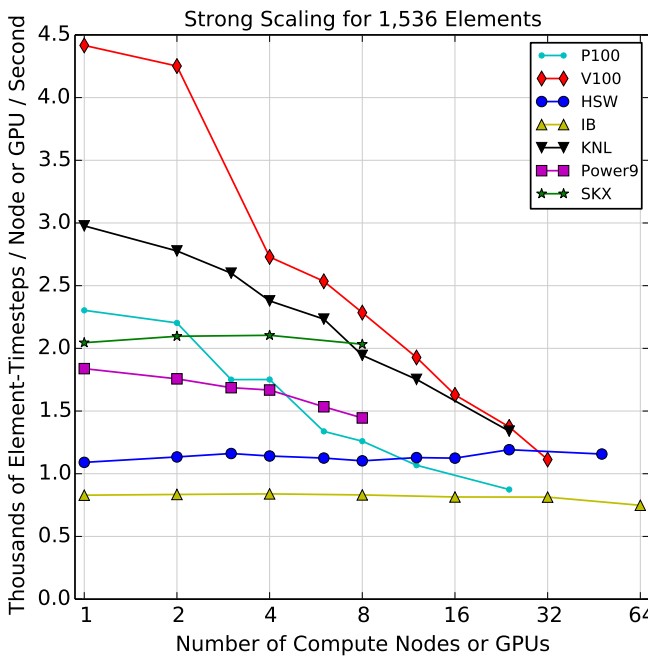

**Figure 11.** Strong scaling at small scale with Intel Xeon Haswell (HSW), Ivy Bridge (IB), and Skylake (SKX); Intel Xeon Phi Knights Landing (KNL); IBM Power9; and Nvidia GPU Pascal (P100) and Volta (V100) architectures for a $1,536$-element grid generated with $n_e = 16$ ($1.875°$ resolution). See caption of Figure 9 for details. Data are for HOMMEXX only.

to compare threading speedup within a KNL core. At 1350 nodes, there is only 1 element/rank; thus, the serial run reveals the OpenMP overhead in HOMMEXX. In HOMME, the outer parallel region has essentially no overhead because it is large. Figure 9(b) omits the Cori-KNL-Serial data for clarity.

Figure 10 reports similar data for the prescribed E3SM version 1 1/4-degree ($n_e = 120$) model. In the version 1 model, (i)
$n_e = 120$; (ii) the number of hyperviscosity subcycles is 4 instead of 3; and (iii) there are 2 horizontal steps per one vertical remapping step, instead of 3. We did not follow this parameter set in this paper because we are measuring data over a large range of $n_e$; to get directly comparable results, we must choose a single set of parameters for all runs.

A run was conducted at 345,600 elements ($n_e = 240$ or $0.125°$ resolution) on up to 5400 Cori KNL nodes. Results are essentially the same as in Figure 9: HOMMEXX's peak efficiency is 2.5, and efficiency drops to 1.2 at 5400 nodes.
After these data were collected, we discovered that HOMME was running a loop that adds zeros to the tracers, as part of source term tendencies that are active only when HOMME is coupled to physics parameterizations. These loops account for approximately 1.5% of the raw HOMME time. To compensate for this discrepancy, in Figures 9 and 10, we removed the maximum time over all ranks for this loop from the reported time. This is a conservative fix in that the timing error favors HOMME rather than HOMMEXX; at worst, too much time is subtracted from HOMME's overall time rather than too little.
These results establish that HOMMEXX achieves at least performance parity with HOMME on NERSC platforms.

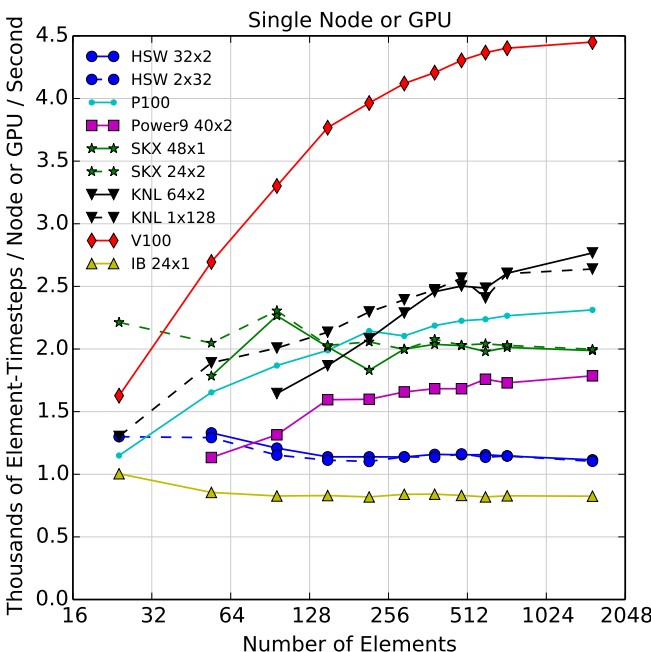

**Figure 12.** Single-node performance as a function of number of elements for several architectures, with various threading options. See caption of Figure 11 for architecture names. Number of MPI ranks $r$ and number of threads $t$ in a configuration is denoted $r \times t$. Data are for HOMMEXX only.

The roughly parabolic shape of the KNL curves is the result of exhausting high-bandwidth memory (HBM) at small node count, and a combination of MPI dominance and on-node efficiency loss at low workload at large node count. Regarding efficiency loss at low workload, see also Figure 12. With $n_q = 40$ tracers, $n_p = 4$, $n_l = 72$ levels, roughly 6 thousand elements fit in the approximately 16 GB of HBM available to the application on a KNL node.

5     Figure 11 studies strong scaling at small scale, with number of elements 1,536, with the same configuration as before but across a broader range of architectures. Here, only HOMMEXX is studied. Again, ideal scaling corresponds to a horizontal line. In both Figures 9 and 11, Intel Xeon efficiency (that is, IB, HSW, and SKX) is roughly flat with increasing node count. In contrast, KNL, GPU, and possibly Power9 efficiency decreases with node count. The effects of the two sockets on GPU performance (visible in the transition from 2 to 3 devices) and then multiple nodes (visible in the transition from 4 to 5 devices)

10    are particularly evident. We anticipate internode communication will be better on Summit than on our testbed.

## 4.2  Single node or device performance

Figures 12 and 13 show results for single-node runs, with 40 tracers as before and also without tracers, respectively. While tracers will always be included in a simulation, far fewer than 40 may be used in some configurations. In addition, a simulation without tracers exposes performance characteristics of the substantially longer and more complicated non-tracer sections of

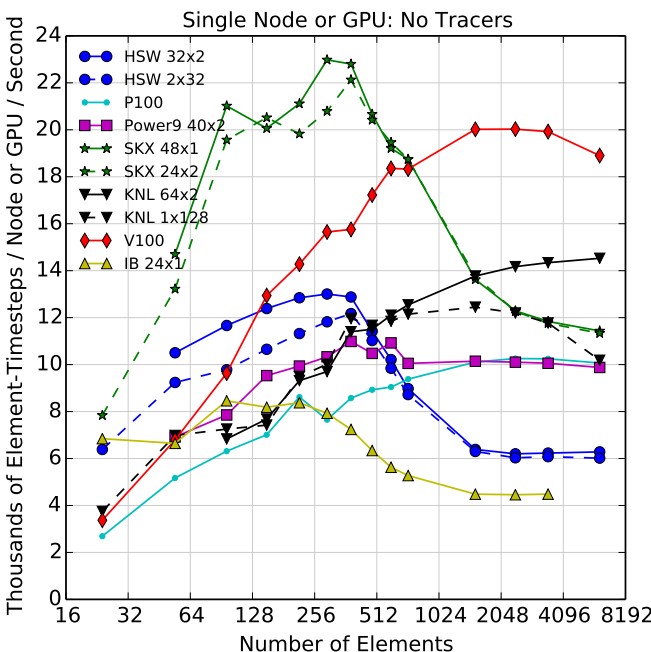

**Figure 13.** As Figure 12, but with no tracers.

code, as well as of the architectures running that code. The no-tracer configuration may be a useful proxy for applications that do not have the benefit of a small section of code devoted to computation over a large number of degrees of freedom.

In the legend, $r \times t$ denotes $r$ ranks with $t$ threads/rank. With tracers, the key result is that on a single node, performance is roughly independent of the separate factors $r$, $t$, depending only on the product $r \cdot t$. Without tracers, a large number of threads

5    per rank does not always perform as well (see, e.g., the drop in performance on KNL when going from a $64 \times 2$ to a $1 \times 128$ configuration), likely because arrays are over elements, and so two arrays used in the same element-level calculation might have data residing on different memory pages.

The runs without tracers (Figure 13) on Intel Xeon CPUs (that is, IB, HSW, and SKX), and also evident to a lesser extent in Figure 12, show that CPU efficiency can vary by as much as a factor of 2 based on workload on a node. This efficiency

10    variation is likely due to the L3 cache. Without tracers, approximately 300 elements can be run on each HSW socket and remain completely in L3. This estimate, doubled because a node has two sockets, coincides with the range of number of elements at which HSW performance drops from just under an efficiency of 13 to approximately 6. If 40 tracers are included, approximately 20 elements/socket fit in L3, which again coincides with the slight drop in HSW efficiency in the case of tracers.

KNL and GPU show a strong dependence on workload, evident in both Figures 12 and 13. For GPU, enough work must

15    be provided to fill all the cores, 2560 in the case of V100, and efficiency is not saturated until each core has multiple parallel loop iterates of work to compute. Nonetheless, the V100 is still reasonably fast even with low workload. For example, at 150 elements and with no tracers, one V100 is faster than one 32-core HSW node. Yet with 150 elements and 16 cores per GPU thread block, only 2400 of the 2560 V100 cores are used, and each for only one loop iterate. However, the Intel Xeon Skylake

is substantially faster than any other architecture at this workload. With tracers, one V100 is always faster than one 32-core HSW node, and one V100 is faster than one 48-core Skylake node at 54 or more elements. We conclude that high workload per GPU is necessary to exploit the GPU's efficiency.

### 4.3 GPU kernel performance

5    We use `nvprof`, an Nvidia performance profiling tool (Nvidia.com, 2018), to assess which important kernels are performing well on GPU and which need improvement. In addition, we compare these kernels on GPU and a 32-core HSW node. From Figure 12, HOMMEXX runs end to end on a V100 device from $1.2\times$ to $3.8\times$ faster than on a HSW node, depending on number of elements, for $n_q = 40$ tracers.

   Table 2 breaks down performance by the major algorithm sets: tracer advection (RK Tracers), dynamics (RK Dynamics), hyperviscosity for dynamics (Hypervis. Dyn.), and vertical remap (Vert. Remap). Within each set, it provides data for up to the three longest-running kernels, listed in descending order of runtime. Data were collected for $n_q = 40$; one V100 device and one HSW node with 32 ranks and 2 threads/rank; and $n_e = 8$ (3.75° resolution) and $n_e = 16$ (1.875° resolution), or 384 and 1,536 elements, respectively. Columns are as follows, where *Alg.* is either a kernel or an algorithm set:

$$p \equiv \frac{\text{Alg. GPU Time}}{\text{HOMMEXX Total GPU Time}}; \quad g \equiv \frac{\text{Alg. CPU Time}}{\text{Alg. GPU Time}}.$$

*Warp eff.* is the ratio of average active threads per warp, the `nvprof` Warp Execution Efficiency metric; and *Occ.* is the 10  percentage of average active warps on a streaming multiprocessor, the Achieved Occupancy metric.

   While most of the kernels within an algorithm set are split up by MPI communication rounds, several of the Vertical Remap and one RK Tracer kernel were split up for profiling. The Hypervis. Dyn. Pre-exchange kernel has a section focused on just the 3 vertical levels near the top boundary; this section leaves some threads idle, leading to low warp efficiency. The RK Dynamics Residual kernel and the Hyperviscosity kernels have low occupancy because of register usage; these are targets for future 15  optimization effort that will focus on increasing the occupancy to 50% through register usage reduction.

### 4.4 Power consumption

Power consumption is an important consideration in new architectures. First we describe power consumption data; Table 1 summarizes these data. Then we discuss how these data may be used to provide another perspective for Figures 9–13.

   The first type of data to consider is manufacturer-specified thermal design power (TDP) for each socket or device. However, 20  node-level power consumption is complicated by other consumers of power. In addition, some devices throttle power. For example, TDP for both the P100 and V100 is 300W, but we observed by continuous sampling by `nvidia-smi` on a single-GPU run that both the P100 and V100 run at approximately 200W for a HOMMEXX workload that fully occupies the GPU. As another example of power throttling, following Grant et al. (2017), the maximum power per HSW node is 415W and per KNL node is 345W. Without frequency throttling, a HSW node was observed in Grant et al. (2017) to run at approximately 25  310 to 355W; without frequency throttling, a KNL node was observed to run at 250–270W. Based on these data, we conclude that 30W should be added to each CPU node to account for other on-node power consumption. For GPU, the host processor's

power must be considered. We were unable to obtain the TDP for Power8 and Power9. We very approximately assign the Power9 node to have 360W. In summary, TDP is not alone sufficient.

In consideration of these complexities, we offer the following numbers as rough guidelines. An Ivy Bridge (IB) node consumes 260W; HSW, 360W; SKX, 330W; KNL, 260W. On Summit, each node will have 6 V100s; thus, we divide the host consumption of very approximately 360W among the 6 V100s for a total of 260W per V100. These numbers are too uncertain to provide an accurate ordinate in our plots. However, the reader may use these numbers to estimate efficiency in terms of power consumption by dividing an efficiency value from a figure by a node type's approximate power consumption; this calculation gives thousands of elements $\times$ time steps per Joule, i.e., number of computations per unit of energy.

## 5 Conclusions

In this work we presented results of our effort to rewrite an existing Fortran-based code for global atmospheric dynamics to a performance portable version in C++. HOMME is a critical part of E3SM, a globally coupled climate model funded by the DOE, and this work was part of the effort to prepare E3SM for future exascale computing resources. Our results show that it is indeed possible to write performance portable code using C++ and Kokkos that can match or exceed the performance of a highly optimized Fortran code on CPU and KNL architectures while also achieving good performance on the GPU.

We presented performance results of the new, end-to-end implementation in HOMMEXX over a range of simulation regimes, and, where possible, compared them against the original Fortran code. Specifically, our results show that HOMMEXX is faster than HOMME on HSW by up to $\sim$1.1$\times$ and faster on KNL by up to $\sim$1.3$\times$. These two results establish that HOMMEXX has better end-to-end performance than a highly tuned, highly used code. Next, in a node-to-device comparison, HOMMEXX is faster on KNL than dual-socket HSW by up to $\sim$2.4$\times$; and HOMMEXX is faster on GPU than on dual-socket HSW by up to $\sim$3.2$\times$. These two sets of results establish that HOMMEXX has high performance on nonconventional architectures, at parity with or in some cases better than the end-to-end GPU performance reported for other codes for similar applications (Demeshko et al., 2018; Howard et al., 2017).

By leveraging the Kokkos programming model, HOMMEXX is able to expose a large amount of parallelism, which is necessary (though not sufficient) for performance on all target architectures. We also provided some details of our implementation, and motivated the choices we made on the basis of portability and performance. In particular, we highlighted a few important design choices that benefited performance on different architectures, such as the change in data layout, thread count-aware sizing of temporary buffers, and the use of explicit vectorization,

Due to architecture differences, in order to achieve portable performance, a substantial effort was needed, including a careful study of the most computationally intensive routines on each architecture, and the adoption of advanced optimization strategies.

The work here is a significant data point to inform strategic decisions on how to prepare E3SM, and other large scientific research code projects, for new HPC architectures. The results are definitive that the approach we took using C++ and Kokkos does lead to performance portable code ready to run on new architectures. We view the results of HOMMEXX's reaching and even exceeding performance parity relative to HOMME on CPU and KNL architectures to be a success, based on our

initial belief that switching to C++ and having the same code base run well on GPUs was going to hurt performance. (In our proposal, our metric of success was to achieve performance portability and to not be more than $15\%$ slower on CPU and KNL.) Furthermore, the results regarding the relative performance of the code across numerous architectures as a function of workload have already been instrumental in planning for scientific runs and computer time allocation requests, since different scientific
inquiries favor maximum strong-scaled throughput and others favor efficiency.

Decisions on whether to write new components, or to pursue targeted rewrites of existing components, in C++ using Kokkos would need to incorporate many other factors beyond the scope of this paper. The objective results here indicate that it is possible to achieve performance in C++ across numerous architectures. It is anticipated, but not yet shown, that a GPU port of the Fortran HOMME code using OpenACC will also be a path to performance on that architecture. In our discussions on the
future programming model for E3SM components, other subjective factors come into play: the availability of code developers and their expertise and preferences, the relative readability and testability of the code, the anticipated portability to future architectures, and the like.

*Code and data availability.*   The source code is publicly available at Bertagna et al. (2018). Performance data in the form of raw timers is available in the same repository in folder `perf-runs/sc18-results` together with build and run scripts.

*Author contributions.*   All authors contributed to the content of the manuscript: All authors contributed to introduction, literature overview and conclusions. L. Bertagna, A. Bradley, M. Deakin, O. Guba, A. Salinger, and D. Sunderland contributed to overview of implementation and performance results. M. Taylor, A. Bradley and O. Guba contributed to the section on HOMME dycore. L. Bertagna, A. Bradley, M. Deakin, O. Guba, and D. Sunderland contributed to code base of HOMMEXX 1.0 and performance runs. Irina K. Tezaur provided infrastructure support and testing. A. Bradley, A. Salinger, M. Taylor, and I. Tezaur provided expertise in high-performance computing, expertise in dycore
modeling, and leadership.

*Competing interests.*   The authors declare that they have no conflict of interest.

*Acknowledgements.*   The authors are grateful to Si Hammond, Kyungjoo Kim, Eric Phipps, Steven Plimpton, and the Kokkos team for their suggestions. The authors thank two anonymous reviewers for their very helpful suggestions. This work was part of the CMDV program, funded by the U.S. Department of Energy (DOE) Office of Biological and Environmental Research. Sandia National Laboratories is a
multimission laboratory managed and operated by the National Technology and Engineering Solutions of Sandia, L.L.C., a wholly owned subsidiary of Honeywell International, Inc., for the DOE's National Nuclear Security Administration under contract DE-NA-0003525. This research used resources of the National Energy Research Scientific Computing Center, a User Facility supported by the Office of Science of DOE under Contract No. DE-AC02-05CH11231. SAND2019-1304 J.

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

**Table 1.** Hardware Summary

| **Machine:** Cori KNL | **Machine:** Cori Haswell |
|---|---|
| • **Architecture:** Knights Landing (KNL)<br>• **Number of nodes:** 9688<br>• **Node configuration:** 68-core Intel Xeon Phi Knights Landing 7250 processor<br>• **Core specifications:** 4 hardware threads, 2 512 bit-wide vector processing units<br>• **Power:** 230 Watts (W) Thermal Design Power (TDP), ∼260 W node<br>• **Memory:** 16 GB MCDRAM with 460 GB/s bandwidth and 96 GB DDR4 2400 MHz memory with 102 GB/s bandwidth<br>• **Notes:** All runs used the quadrant NUMA mode and the cache memory mode. Located at NERSC. | • **Architecture:** Haswell (HSW)<br>• **Number of nodes:** 2388<br>• **Node configuration:** 2 sockets, each is a 16-core Intel Xeon E5-2698 v3 Haswell processor<br>• **Core specifications:** 2 hardware threads, 2 256 bit-wide vector processing units<br>• **Power:** 330 W TDP, ∼360 W node<br>• **Memory:** 40 MB L3 cache per socket, 128 GB DDR4 2133 MHz per node<br>• **Notes:** Located at NERSC. |
| **Machine:** Edison | **Machine:** Skylake testbed |
| • **Architecture:** Ivy Bridge (IB)<br>• **Number of nodes:** 5586<br>• **Node configuration:** 2 sockets, each is a 12-core Intel Xeon E5-2695 v2 Ivy Bridge processor<br>• **Core specifications:** 2 hardware threads, 1 256 bit-wide vector processing unit<br>• **Power:** 230 W TDP, ∼260 W node<br>• **Memory:** 30 MB L3 cache per socket, 64 GB DDR3 1866 MHz per node<br>• **Notes:** Located at NERSC. | • **Architecture:** Intel Xeon Skylake (SKX)<br>• **Node configuration:** 2 sockets, each is a 24-core Intel Xeon Platinum 8160 processor<br>• **Core specifications:** 2 hardware threads, 2 512 bit-wide vector processing units<br>• **Power:** 300 W TDP, ∼330 W node<br>• **Memory:** 32 MB L3 cache per socket, 196 GB DDR4 2666 MHz per node<br>• **Notes:** Faster memory relative to HSW, has 6 memory channels relative to HSW's 4. Located at SNL. |
| **Machine:** V100 testbed | **Machine:** P100 testbed |
| • **Architecture:** Power9 CPU, Nvidia Volta V100 GPU<br>• **Node configuration:** 2 sockets, each is a 20-core IBM Power9 processor with 2 V100 GPUs<br>• **Core specifications:** Power9 core has 4 hardware threads; each V100 has 2560 double-precision cores, with 32 threads/core<br>• **Power:** V100 has 300 W TDP; Power9 TDP is unknown; V100 runs at ∼200 W<br>• **Notes:** The GCC 7.20 compiler and CUDA 9.1.85 were used. One MPI rank was assigned to each GPU. Located at SNL. | • **Architecture:** Power8 CPU, Nvidia Pascal P100 GPU<br>• **Node configuration:** 2 sockets, each is an 8-core Power8 Firestone processor with 2 P100 GPUs<br>• **Core specifications:** Power8 core has 8 hardware threads; P100 has 1792 double-precision cores, with 32 threads/core<br>• **Power:** P100 has 300 W TDP; Power8 TDP is unknown; P100 runs at ∼200 W<br>• **Notes:** The GCC 5.4.0 compiler and CUDA 8.0.44 were used. One MPI rank was assigned to each GPU. Located at SNL. |

**Table 2.** V100 GPU Performance Study

| | 384 elems. | | | | 1536 elems. | |
|---|---|---|---|---|---|---|
| Kernel | $p$ | $g$ | Warp Eff. (%) | Occ. (%) | $p$ | $g$ |
| Homme Total | 1.0 | 3.7 | —— | —— | 1.0 | 4.2 |
| RK Tracers | 0.69 | 4.5 | —— | —— | 0.73 | 4.5 |
| Advection | 0.13 | 2.3 | 92 | 62 | 0.14 | 2.4 |
| Limiter | 0.093 | 5.8 | 99 | 55 | 0.1 | 5.3 |
| Pre-exchange | 0.065 | 2.1 | 91 | 62 | 0.07 | 2.1 |
| RK Dynamics | 0.13 | 1.8 | —— | —— | 0.11 | 3.7 |
| Residual | 0.085 | 1.3 | 92 | 47 | 0.077 | 2.0 |
| Hypervis. Dyn. | 0.11 | 1.1 | —— | —— | 0.091 | 3.9 |
| Laplacian 1 | 0.021 | 1.6 | 91 | 23 | 0.019 | 3.1 |
| Laplacian 2 | 0.022 | 1.5 | 91 | 33 | 0.021 | 2.1 |
| Pre-exchange | 0.018 | 0.83 | 53 | 24 | 0.015 | 2.8 |
| Vert. Remap | 0.053 | 4.3 | —— | —— | 0.055 | 4.6 |
| PPM | 0.018 | 4.0 | 64 | 46 | 0.019 | 3.9 |
| Remap | 0.02 | 1.9 | 62 | 24 | 0.021 | 2.0 |
| Cell Means | 0.0094 | 6.0 | 86 | 46 | 0.01 | 6.4 |