# Peer review of "HOMMEXX 1.0: A Performance Portable Atmospheric Dynamical Core for the Energy Exascale Earth System Model"

_Geoscientific Model Development, 2018_

## Referee Comment (RC1) · Anonymous Referee #1 · 23 Nov 2018

Summary The authors shared their experiences and results in porting the dynamical core model (HOMME) of an Earth System Model (E3SM) from optimized Fortran implementation to C++ with the Kokkos library, with the aim to achieve performance portability on conventional CPUs, KNL and GPUs. The experimental results show that the new implementation (HOMMEXX) matches the performance and strong-scaling characteristics of the Fortran code on Haswell CPU cluster, and improves the performance on KNL. The abstraction of parallel loops in Kokkos opens the possibility of porting to GPUs, where the results show better performance on V100 compared with a Haswell node.

[Figure]

Comments - I think the introduction and problem description is clear for someone from other fields in numerical methods to follow without too much difficulty. For completeness, the authors could consider adding mathematical equations for the differential operators. - I feel Section 3.3 needs some improvement. I found the part describing how paralle_for loops map to different execution policies slightly unclear. I suggest describing the kernel with pseudo-code of nested loops, decorated with execution policy choice. - Have the authors verified that vectorization on CPU is effective, potentially by looking at the generated assembly code? - I'm curious that if the authors encountered any limitations for the vector data types, e.g. for maths function calls, conditionals etc. - Could the authors elaborate more on "reuse of subviews is important to minimize index arithmetic" on CPU (page 11 line 9)? I don't quite follow what is "... the number of connections per elements..." (page 11 line 12). - In Section 4, HOMME and HOMMEXX have very similar performance on Haswell, but using different (if I understand correctly) strategy, could the authors explore a bit more on the reason behind it? - Subtle point: is Turbo-Boost a potential source of randomness in the experiments? - One thing I feel the paper is missing is that we do not know if the achieved performance is "good enough". The paper could be improved (by a lot) by e.g. showing the percentage of peak performance achieved and/or roofline model of the hardware. This is especially helpful because the experiments are carried on a large range of hardware with very different characteristics, and finding some common metrics to compare and contrast between them would help the authors in organizing the presentation of experimental results.

---

## Referee Comment (RC2) · Anonymous Referee #2 · 29 Nov 2018

"HOMMEXX 1.0: A Performance Portable Atmospheric Dynamical Core for the Energy Exascale Earth System Model"

Authors: Luca Bertagna, Michael Deakin, Oksana Guba, Daniel Sunderland, Andrew M. Bradley, Irina K. Tezaur, Mark A. Taylor, and Andrew G. Salinger

The authors describe porting the HOMME dynamical core (used by the atmospheric component in both E3SM and CESM) to HOMMEXX. HOMMEXX is a single source code thats supports execution on both CPUs and GPUs. The work required to transform this sizable code base to be (mostly) platform independent was likely quite substantial and was a rather bold undertaking. Because climate simulations are performed

by the community on a variety of architectures, the single code base solution is appealing - especially in preparation for exascale computing. While HOMMEXX may not get the performance on GPU that one would aspire to, it's frankly impressive that the code transformation did not result in worse performance on the CPU. Also I appreciate that many architectures were evaluated. That said, I do think that the manuscript itself could use some improvements, which I detail below.

—————————————- Comments/suggestions/questions —————————————-

(1) The authors do say that there was substantial work involved in this refactoring, but I am interested in them quantifying it in some way (if possible) - e.g., people hours, or lines of code touched, or percentage of HOMME code that is unchanged in HOMMEXX ?

(2) On the same line of thinking, can the authors speculate how this effort would compare to the effort that would be required to use OpenACC to port HOMME to GPUs. (Section 5 indicates that the authors are considering this undertaking as well.)

(3) I would have liked to see some performance metrics in the paper that are in more common use by the climate modeling community, e.g. simulated years per compute day or CPU-core hours per simulation year.

(4) Because the stated motivation of this work is running at exascale, it seems that the Dennis et al 2017 paper on this subject ("Preparing the Community Earth System Model for Exascale Computing") should probably be cited and mentioned in Section 1.

(5) Page 4: text refers to Figure 4 before Figure 3 (swap order of these figures?)

(6) When referring to a resolution for the first time in the text via n_e (e.g., n_e = 240), please also mention the corresponding grid degree (as done for n_e = 30 on page 5 and in Fig. 4 caption).

(7) Section 2: Why only use the outer threading for HOMME? Please explain/justify.

(8) In general, I'd like to see more detail given in Section 3, given that this is a journal paper (as opposed to length-constrained C.S. conference submission) that should be of interest to climate modelers. It may be necessary to break up section 3 if significant more detail is added.

–Section 3.1: I really would like more justification for the choice of using Kokkos. And I'd like to understand more of what was involved to use it - how about a simple example?

–Section 3.2: What is the correctness-testing build? (p.8. line 15). Is this what lines 16-17 are describing? I assume that this is not what you use for performance testing? Please clarify and consider expanding the correctness discussion.

– Section 3.3: this section could benefit from some code snipets/pseudocode to clarify

(9) page 11, line 9: What is a subview?

(10) I believe that Section 4 (results) could be improved quite a bit. It was a bit frustrating at times:

–section 4.1: I'd like it to be easier to get the info on the various test machines and figure out which was which in the figures - so maybe put them in a bulleted list or large table. I'd suggest that the name of the platform - as referred to in the figures should be in bold and appear first. For those readers not familiar with these DOE machines, it was frustrating that Fig. 5 listed "Edison", which was alternatively referred to by its processor (IB) in Figure 6. I kept having to search through information in paragraphs on pages 12 and 13 while looking at the plots. Similarly in section 4.3 (page 16, line 17), when a reference was made to the Xeon machines as a group, I had to search through these paragraphs. I think that not all readers are familiar with these machines, so please make it more accessible.

–page 14: this discussion of power consumption feels stuck in here. The rough guideline numbers on power use (lines 10-14) should probably be included in the bulleted list or table of machines. Or maybe this power discussion should be in a new section

4.5 that happens after the reader has seen the other results - making it easier to follow. As it is, figures 5-8 are referred to in line 2 before they have really been presented in the text (which happens in later subsections).

–page 15, lines 10-11: this info feels like it should be in the intro of section 4 (page 12) - not stuck at the end of 4.1

–section 4.1: I also find it unhelpful to refer to the figures before "presenting" them. For example, page 15, line 1 and page 15, line 3: Theses comments about the plot attributes should go in the sections where the plots are described (4.2 and 4.3)

–Consider combining 4.2 and 4.3 into a single "strong scaling" section. Also make the machine labels more consistent between plots 5 and 6, for example. (Even though 5 has fewer platforms.)

–Section 4.4: Overall, interpreting the results could be easier ( more readable) by referring in the text to specific examples in the figures. For example, in page 16 line 29, say which platform in Figure 8 is the one that "does not always perform as well" with a large number of threads per rank.

(11) Table 1: consider naming the kernels for those familiar with HOMME (rather than kernel 1, kernel 2, ...)

(12) section 4.5, lines 14-15: It seems that Figure 6 does not indicate that V100 is strictly faster than HSW, though this text suggests that (the 1.2x to 3.8x)

(13) page 13, line 8: I don't know whether to be concerned about what else may be hardware-specific in this single source code. What percentage of code is different? What are the types of code constructs that are problematic for Kokkos? What are the broader implications for other codes?

————————————- Minor items: ————————————-

(1) page 7, line 6: "around" => "on"

(2) page 9, line16: "the the" => "the"

(3) page 9, line 19: "nested loops," => "nested loops"

(4) page 17, line 12: I'd assume that the reader is not necessarily aware of what nvprof is...should at least cite this.

---

## Author Response (AR1)

**HOMMEXX 1.0: A Performance Portable Atmospheric Dynamical Core for the Energy Exascale Earth System Model**

Luca Bertagna[1], Michael Deakin[1], Oksana Guba[1], Daniel Sunderland[1], Andrew M. Bradley[1], Irina K. Tezaur[1], Mark A. Taylor[1], and Andrew G. Salinger[1]

[1]Sandia National Laboratories, PO Box 5800, Albuquerque, NM, 87175 USA

**Correspondence:** Luca Bertagna (lbertag@sandia.gov)

**Abstract.** We present an architecture-portable and performant implementation of the atmospheric dynamical core (HOMME) of the Energy Exascale Earth System Model (E3SM). The original Fortran implementation is highly performant and scalable on conventional architectures using MPI and OpenMP. We rewrite the model in C++ and use the Kokkos library to express on-node parallelism in a largely architecture-independent implementation. Kokkos provides an abstraction of a compute node or device, layout-polymorphic multidimensional arrays, and parallel execution constructs. The new implementation achieves the same or better performance on conventional multicore computers and is portable to GPUs. We present performance data for the original and new implementations on multiple platforms, on up to 5400 compute nodes, and study several aspects of the single- and multi-node performance characteristics of the new implementation on conventional CPU, Intel Xeon Phi Knights Landing, and Nvidia V100 GPU.

*Copyright statement.* HOMMEXX version 1.0: Copyright 2018 National Technology & Engineering Solutions of Sandia, LLC (NTESS). Under the terms of Contract DE-NA0003525 with NTESS. For full copyright statement, see **?** .

[revised manuscript text omitted]

horizontal step

5 RK stages
for dynamics

3 HV subcycles
for dynamics HV

3 RK stages
for tracers and HV

total time step in HOMME

3 horizontal steps

vertical
remap

**Figure 4.** HOMME timestepping configuration.

160 ## 3 Overview of the implementation

**3.1 The Kokkos library**

 We achieve performance portability of HOMME  by using Kokkos. Kokkos is a C++11 library and programming model that enables developers to write performance portable thread-parallel codes on a wide variety of HPC architectures (**?**). Kokkos is used to optimize on-node performance, allowing HPC codes to leverage their existing strategies for optimizing inter-node performance. Here, we briefly describe the key premises on which Kokkos is based.

Kokkos exposes several key compile time abstractions for parallel execution and data management which help its users to develop performance portable code. The following execution abstractions are used to describe the parallel work.

- A **kernel** is a user provided body of work that is to be executed in parallel over a collection of user defined work items. Kernels are required to be free of data dependencies, so that a kernel can be applied to the work items concurrently without an ordering.

- An **execution space** describes where a kernel should execute; for example, whether a kernel should run on the GPU or the CPU.

- An **execution pattern** describes how a kernel should run in parallel. Common execution patterns are `parallel_for`, `parallel_reduce`, and `parallel_scan`.

- The **execution policy** describes how a kernel will receive work items. A **range** execution policy is created with a pair of lower (L) and upper (U) bounds, and will invoke the kernel with an integer argument for all integers i in the interval [L, U).  A **team** execution policy is created with a number of teams and number of threads per team. The strength of a team policy is that threads within a team can cooperate to perform shared work, allowing for additional levels of parallelism to be exposed within a kernel. Team policies allow users to specify up to three levels of hierarchical parallelism: over the number of teams, the number of threads in a team, and the vector lanes of a thread.

Kokkos also provides memory abstractions for data. The main one is a multidimensional array reference which Kokkos calls `View`. `View`s have four key abstractions: (1) data type, which specifies the type of data stored in the `View`; (2) layout, which describes how the data is mapped to memory; (3) memory space, which specifies where the data lives; and (4) memory traits, which indicates how the data should be accessed. The `View` abstractions allow developers to code and verify algorithms one time, while still leaving  flexibility in the memory layout, such as transposing the underlying data layouts for CPU versus GPU architectures. Although `View`s are used to represent N-dimensional arrays, internally Kokkos `View`s store a one-dimensional array. When accessing an element (identified by a set of N indices), Kokkos performs integer arithmetic to map the input indices to a position in the internal one-dimensional array. As we will discuss in section**??**, this fact impacted our design choices.

Kokkos uses C++ template metaprogramming to specify the instantiations of the execution space and data/memory abstractions of data objects and parallel  execution constructs, to best optimize code for the specified HPC architecture. This allows users to write on-node parallel code that is portable and can achieve high performance.

```
\DIFaddFL{double a}[\DIFaddFL{N0}][\DIFaddFL{N1}][\DIFaddFL{N2}]\DIFaddFL{;
double b}[\DIFaddFL{N0}][\DIFaddFL{N1}][\DIFaddFL{N2}]\DIFaddFL{;
for (int i=0; i<N0; ++i) }{
  \DIFaddFL{for (int j=0; j<N1; ++j) }{
    \DIFaddFL{for (int k=0; k<N2; ++k) }{
      \DIFaddFL{a(i,j,k) = some_function(b(i,j,k));
}}}}
```

```
\DIFaddFL{Kokkos::View<
Kokkos::View<double}[\D
Kokkos::parallel_for (
  Kokkos::RangePolicy<K
is achieved through ca
\DIFdelFL{, }\texttt{\D
\DIFdelFL{, and }\text
\DIFdelFL{templated fu
  }[\DIFaddFL{=}]\DIFa
    \DIFaddFL{int i = i
    int j = idx / N2, l
    a(i,j,k) = some_fu
}}\DIFaddFL{);
}
```

$\Longrightarrow$

**Figure 5.** Parallelization of tightly nested loops via flattening and range policy. After flattening, modular arithmetic is used by each thread to determine the portion of input and output arrays to operate on.

HOMMEXX  uses Kokkos `Views` for data management and Kokkos execution patterns for intra-MPI-rank parallelism. Hierarchical parallelism is achieved with team policies and is crucial in HOMMEXX (see section **??**).

To give an example of Kokkos syntax, we briefly present two simple pseudocode examples. Figure **??** shows how a series of tightly nested for loops can be translated to a single flattened for loop, exposing maximum parallelism. The resulting single for loop can be parallelized with Kokkos, using a simple range policy; this procedure corresponds to using OpenMP's *collapse* clause. Figure **??** shows a different scenario, in which the output of a matrix-vector multiplication (with multiple right hand sides) is used as input for a second matrix-vector multiplication (again, with multiple right hand sides). Since the second multiplication depends on the output of the first, the nested for loops cannot be flattened as in the previous case. In order to use a range policy, one would have to dispatch the parallel for only on the outermost loop, which would fail to expose all the parallelism (since all the rows in each of the two multiplications can in principle be computed in parallel). With a team policy, on the other hand, threads are grouped to form teams; teams act on the same outer iteration, can share temporary variables, and can be used to further parallelize inner loops. In this example, a team is assigned to one right hand side, and each member of the team is assigned a subset of the rows. Notice that, since the second multiplication cannot start until the first one is fully completed, a team synchronization is necessary.

Kokkos is just one of the possible ways in which one can achieve performance portability. In particular, we can identify three approaches:

– Compiler directives: In this approach, preprocessor directives expose parallelism. Included in the directives are instructions to target a specific architecture. Examples include OpenACC (**?**) and OpenMP (**?**). This strategy has the advantage of

```
\DIFaddFL{constexpr int NUM_RHS = 9;
constexpr int N = 32;
double A}[\DIFaddFL{N}][\DIFaddFL{N}]\DIFaddFL{;
double B}[\DIFaddFL{N}][\DIFaddFL{N}]\DIFaddFL{;
double x}[\DIFaddFL{NUM_RHS}][\DIFaddFL{N}]\DIFaddFL{;
double y}[\DIFaddFL{NUM_RHS}][\DIFaddFL{N}]\DIFaddFL{;
double z}[\DIFaddFL{NUM_RHS}][\DIFaddFL{N}]\DIFaddFL{;
// Initialize arrays }[\DIFaddFL{...omitted...}]
\DIFaddFL{for (int i=0; i<NUM_RHS; ++i) }{
  \DIFaddFL{for (int j=0; j<N; ++j) }{
    \DIFaddFL{for (int k=0; k<N; ++k) }{
      \DIFaddFL{y}[\DIFaddFL{i}][\DIFaddFL{j}] \DIFaddFL{+= A}[\DIFaddFL{j}][\DIFaddFL{k}]\DIFaddFL{*x}[\DIFaddFL{
    }
  }
  \DIFaddFL{for (int j=0; j<N; ++j) }{
    \DIFaddFL{for (int k=0; k<N; ++k) }{
      \DIFaddFL{z}[\DIFaddFL{i}][\DIFaddFL{j}] \DIFaddFL{+= B}[\DIFaddFL{j}][\DIFaddFL{k}]\DIFaddFL{*y}[\DIFaddFL{
    }
  }
}
```

[revised manuscript text omitted]

                    const ExecViewUnmanaged<const Scalar }[\DIFaddFL{2}][\DIFaddFL{NP}][\DIFaddFL{NP}][\DIFaddFL{NUM
                    const ExecViewUnmanaged<       Scalar      }[\DIFaddFL{NP}][\DIFaddFL{NP}][\DIFaddFL{NUM_LEV}]\DIFa
  \DIFaddFL{const auto}& \DIFaddFL{Dinv = Homme::subview(m_dinv,kv.ie);
  const auto}& \DIFaddFL{metdet = Homme::subview(m_metdet,kv.ie);
  const auto}& \DIFaddFL{buf = Homme::subview(vector_buf_ml,kv.team_idx,0);
  Kokkos::parallel_for(Kokkos::TeamThreadRange(kv.team, NP*NP), }[&]\DIFaddFL{(const int loop_idx) }{
    \DIFaddFL{const int igp = loop_idx / NP, jgp = loop_idx %DIF > NP;
    Kokkos::parallel_for(Kokkos::ThreadVectorRange(kv.team, NUM_LEV), }[&] \DIFaddFL{(const int}& \DIFaddFL{ilev) }
      \DIFaddFL{const auto}& \DIFaddFL{v0 = v(0, igp, jgp, ilev);

[revised manuscript text omitted]

**Responses to reviewers**

**Responses to reviewer 1**

We thank the reviewer for their comments and remarks. Here, we summarize how we addressed their comments.

1) RC: I think the introduction and problem description is clear for someone from other fields in numerical methods to follow without too much difficulty. For completeness, the authors could consider adding mathematical equations for the differential operators.

AC: Equations that describe HOMME are added at the beginning of the HOMME section.

2) RC: I feel Section 3.3 needs some improvement. I found the part describing how parallel_for loops map to different execution policies slightly unclear. I suggest describing the kernel with pseudo-code of nested loops, decorated with execution policy choice.

AC: We added one generic example in section 3.1, and one example specific to HOMME in section 3.4.2.

3) RC: Have the authors verified that vectorization on CPU is effective, potentially by looking at the generated assembly code?

AC: We have. To explain our approach, we have expanded the material on vectorization to explain our analysis and results. We have provided a new figure to show performance as a function of the pack length and operator implementation approach.

4) RC: I'm curious that if the authors encountered any limitations for the vector data types, e.g. for maths function calls, conditionals etc.

AC: We added some sentences about conditionals. HOMME has very few low-level conditionals, unlike, for example, physics parameterizations.

5) RC: Could the authors elaborate more on "reuse of subviews is important to minimize index arithmetic" on CPU (page 11 line 9)? I don't quite follow what is "... the number of connections per elements..." (page 11 line 12).

AC: We expanded the sentence in 3.4.1 to explain what we meant by that.

6) RC: In Section 4, HOMME and HOMMEXX have very similar performance on Haswell, but using different (if I understand correctly) strategy, could the authors explore a bit more on the reason behind it?

AC: On conventional CPU, such as Haswell, and especially in the serial build (1 thread/core, 1 MPI rank/core), the two implementations are quite similar. In this case, our goal is to match HOMME's performance, with the primary challenges being (i) matching HOMME's excellent auto-vectorization and (ii) not slowing down this run configuration due to strategies used for other architectures.

7) RC: Subtle point: is Turbo-Boost a potential source of randomness in the experiments?

AC: Turbo-Boost and other throttling in either direction, such as decreasing the clock speed on Skylake when AVX512 instructions are being run, is unlikely to be any more a source of randomness than other effects, such as network physical topology. Indeed, experience on supercomputers is that the interconnect tends to be the greatest source of run-to-run variability. To minimize the impact on code comparability of noise effects in general, side-by-side comparison was done always in the same job submission, so exactly the same computer nodes are used within a temporally constrained period of time to obtain comparison results. To clarify this point, a sentence on jobs submission was added at the beginning of section 4.

8) RC: One thing I feel the paper is missing is that we do not know if the achieved performance is "good enough". The paper could be improved (by a lot) by e.g. showing the percentage of peak performance achieved and/or roofline model of the hardware. This is especially helpful because the experiments are carried on a large range of hardware with very different characteristics, and finding some common metrics to compare and contrast between them would help the authors in organizing the presentation of experimental results.

AC: First, for clarity, we would like to emphasize that the metric of "(thousands of elements-timesteps)/(node or GPU)/second", while complicated, is a useful efficiency metric. Data in these units can be directly compared across architectures (e.g., GPU vs KNL vs HSW vs SKX) and across scaling regimes (many elements/compute resource vs. few elements/resource). Second, we agree that careful systems-oriented metrics would be interesting. These would characterize inter-node bandwidth and latency; on-node bandwidth, cache performance, GPU kernel launch latency, CPU-GPU bandwidth, and many other GPU metrics. However, any set of metrics requires a lot of work to collect and analyze. We have chosen to prioritize the key metrics of interest: 1. Does HOMMEXX match HOMME wherever HOMME can run? 2. Does HOMMEXX vectorize well? 3. Across scaling

regimes, both isolating on-node performance and accounting also for inter-node communication, how does the the end-to-end performance vary? 4. What is the performance on nonconventional archictectures relative to conventional ones? 1, 3, and 4 are already answered in the original manuscript, and we have improved the exploration of question 2 in the revised manuscript, in response to thoughtful questions regarding vectorization. That said, the substance of this question essentially points to the research topic of speeding up the dycore independently of implementation strategy. That is, can we isolate an important section of code, collect careful systems data, and use it to speed it up, even just in the original HOMME Fortran code? This is a great question. We do not attempt to answer that question in this paper.

**Responses to reviewer 2**

We thank the reviewer for their comments and remarks. Here, we summarize how we addressed their comments.

1) RC: The authors do say that there was substantial work involved in this refactoring, but I am interested in them quantifying it in some way (if possible) - e.g., people hours, or lines of code touched, or percentage of HOMME code that is unchanged in HOMMEXX ?

AC: We estimated the effort in terms of lines of code. In the introduction, we added this text: "This effort produced 13,000 new lines of code in C++ and 2,000 new lines of code in Fortran. Most of these lines replace code that solves the dynamical and tracer equations in HOMME. HOMMEXX uses HOMME's Fortran initialization routines."

2) RC: On the same line of thinking, can the authors speculate how this effort would compare to the effort that would be required to use OpenACC to port HOMME to GPUs. (Section 5 indicates that the authors are considering this undertaking as well.)

AC: We extended section 3.1 to give a better idea of where Kokkos stands in the broader context of performance portability. We listed other possible approaches, highlighting features of the different approaches. We believe this should help the reader to better understand how the current approach compares with others.

3) RC: I would have liked to see some performance metrics in the paper that are in more common use by the climate modeling community, e.g. simulated years per compute day or CPU-core hours per simulation year.

AC: We now include SYPD figures for the case of ne=120. For the other figures, we like to use a metric that is invariant to the number of elements in the mesh.

4) RC: Because the stated motivation of this work is running at exascale, it seems that the Dennis et al 2017 paper on this subject ("Preparing the Community Earth System Model for Exascale Computing") should probably be cited and mentioned in Section 1.

AC: Thank you for pointing out this very relevant reference; we added it to the introduction.

5) RC: Page 4: text refers to Figure 4 before Figure 3 (swap order of these figures?)

AC: Thank you, a few figures were re-ordered. We are planning to adjust the position of the figures once the manuscript is ready for its final typesetting.

6) RC: When referring to a resolution for the first time in the text via $n_e$ (e.g., $n_e = 240$), please also mention the corresponding grid degree (as done for $n_e = 30$ on page 5 and in Fig. 4 caption).

AC: Throughout the text, we added resolutions in degrees next to each $n_e$ parameter.

7) RC: Section 2: Why only use the outer threading for HOMME? Please explain/justify.

AC: We edited the text as follows: "On conventional CPUs, HOMME supports outer OpenMP threading over elements and inner OpenMP threading over vertical levels and tracers. The outer threads are dispatched at the driver's top-level loop in one large parallel region. Each type of threading can be turned on or off at configure time, and, if both are on, they lead to nested OpenMP regions. If the number of elements in a rank is at least as large as the number of hardware threads associated with the rank, then it is best to enable only the outer threads. For this reason, in our performance comparisons we configure HOMME with outer threads only, unless stated otherwise."

8) RC: In general, I'd like to see more detail given in Section 3, given that this is a journal paper (as opposed to length-constrained C.S. conference submission) that should be of interest to climate modelers. It may be necessary to break up section 3 if significant more detail is added.

AC: In section 3.2 we highlighted in bold where a new design choice starts.

a) Section 3.1: I really would like more justification for the choice of using Kokkos. And I'd like to understand more of what was involved to use it - how about a simple example?

AC: In section 3.4.2 we added a code snippet showing how a common kernel (the divergence operator) is implemented in HOMMEXX using Kokkos. This snippet also highlights the three design choices mentioned in section 3.3.

b) Section 3.2: What is the correctness-testing build? (p.8. line 15). Is this what lines 16-17 are describing? I assume that this is not what you use for performance testing? Please clarify and consider expanding the correctness discussion.

AC: We expanded the paragraph a little bit, to help clarify what we meant with the concept of correctness-testing build.

c) Section 3.3: this section could benefit from some code snipets/pseudocode to clarify

AC: See answer to point a).

9) RC: page 11, line 9: What is a subview?

AC: We expanded the sentence to better explain the concept.

10) RC: I believe that Section 4 (results) could be improved quite a bit. It was a bit frustrating at times:

a) section 4.1: I'd like it to be easier to get the info on the various test machines and figure out which was which in the figures - so maybe put them in a bulleted list or large table. I'd suggest that the name of the platform - as referred to in the figures should be in bold and appear first.

AC: We added Table 1, Hardware Summary, to describe each machine/architecture.

b) For those readers not familiar with these DOE machines, it was frustrating that Fig. 5 listed "Edison", which was alternatively referred to by its processor (IB) in Figure 6. I kept having to search through information in paragraphs on pages 12 and 13 while looking at the plots. Similarly in section 4.3 (page 16, line 17), when a reference was made to the Xeon machines as a group, I had to search through these paragraphs. I think that not all readers are familiar with these machines, so please make it more accessible.

AC: We changed the name to "Edison-IB" in the figure to connect the platform used in the large-scale runs with the architecture also studied in the single-node runs.

c) page 14: this discussion of power consumption feels stuck in here. The rough guideline numbers on power use (lines 10-14) should probably be included in the bulleted list or table of machines. Or maybe this power discussion should be in a new section 4.5 that happens after the reader has seen the other results - making it easier to follow. As it is, figures 5-8 are referred to in line 2 before they have really been presented in the text (which happens in later subsections).

AC: We moved the discussion on power consumption to a separate subsection, We also included the power consumption of each machine in Table 1. Power consumption is a complicated issue; in the absence of systems-level instrumentation, to which we lack access, we must make careful inferences. We are sure to be clear about this uncertainty in the text.

d) page 15, lines 10-11: this info feels like it should be in the intro of section 4 (page 12) - not stuck at the end of 4.1

AC: As suggested, we moved it at the beginning of section 4.

e) section 4.1: I also find it unhelpful to refer to the figures before "presenting" them. For example, page 15, line 1 and page 15, line 3: Theses comments about the plot attributes should go in the sections where the plots are described (4.2 and 4.3)

AC: Thanks. We are planning to adjust the position of the figures once the manuscript is ready for its final typesetting.

f) Consider combining 4.2 and 4.3 into a single "strong scaling" section. Also make the machine labels more consistent between plots 5 and 6, for example. (Even though 5 has fewer platforms.)

AC: We fused section 4.2 and 4.3 into a single strong scaling section.

g) Section 4.4: Overall, interpreting the results could be easier ( more readable) by referring in the text to specific examples in the figures. For example, in page 16 line 29, say which platform in Figure 8 is the one that "does not always perform as well" with a large number of threads per rank.

AC: Thanks. We highlighted the architecture name in a few places where the generic Intel Xeon name was used, or where no particular architecture was named (as in the example mentioned by the reviewer).

11) RC: Table 1: consider naming the kernels for those familiar with HOMME (rather than kernel 1, kernel 2, ...)

AC: Thanks; done.

12) RC: section 4.5, lines 14-15: It seems that Figure 6 does not indicate that V100 is strictly faster than HSW, though this text suggests that (the 1.2x to 3.8x)

AC: Thanks. We revised this sentence to refer to Figure 7 only. Figure 6 includes inter-node communication on a testbed; the intent of this sentence is to summarize on-node/device performance, in this case comparing 1 V100 device with 1 HSW node.

13) RC: page 13, line 8: I don't know whether to be concerned about what else may be hardware-specific in this single source code. What percentage of code is different? What are the types of code constructs that are problematic for Kokkos? What are the broader implications for other codes?

AC: We added the text: "We also use architecture-dependent code for three vertical integrals in the dynamics and the the tracer transport limiter. The total number of architecture-specific lines of code is approximately 800, out of approximately 13,000 C++ lines of code."

14) RC: (1) page 7, line 6: "around" => "on" (2) page 9, line16: "the the" => "the" (3) page 9, line 19: "nested loops," => "nested loops" (4) page 17, line 12: I'd assume that the reader is not necessarily aware of what nvprof is...should at least cite this.

AC: Thank you. Typos were fixed, and a reference to nvprof was added.